# Directly synthesized cobalt oxyhydroxide as an oxygen evolution catalyst in proton exchange membrane water electrolyzers

Jinzhen Huang [1] ✉, Zheyu Zhang [1], Chiara Spezzati[2], Adam H. Clark [3], Natasha Hales [1], Nina S. Genz [3], Niéli Daffé [3], Radim Skoupy [4], Lorenz Gubler [1], Ivano E. Castelli [2], Thomas J. Schmidt [1,5] & Emiliana Fabbri [1] ✉

The limited choice of oxygen evolution reaction catalysts for proton exchange membrane water electrolyzers hinders their large-scale commercialization. Cobalt-based catalysts are promising candidates and usually undergo surface reconstruction into CoOOH-like structures. However, the directly synthesized CoOOH has not yet been investigated in acidic environments. Here, we show that the CoOOH is active across the whole pH range, while its redox features are pH dependent. Operando hard X-ray absorption spectroscopy characterizations show a pH-induced change in Co oxidation onset, but no change in the coverage of redox-active Co species before the oxygen evolution reaction. The pH-dependent catalytic performance is connected to the interfacial Co oxidative transformations under electrocatalytic conditions. By combining the kinetic isotope effect and the apparent activation energy with theoretical verification, we offer the mechanistic discussion of the possible reaction pathway for CoOOH. In addition, CoOOH demonstrates a stable cell potential of 100 mA cm$^{-2}$ for 400 h in a proton exchange membrane water electrolyzer. These results shed light on both the fundamental electrochemical properties of CoOOH and its potential for practical device applications.

A transition to net-zero $CO_2$ emissions depends on the availability of a vast amount of inexpensive and clean electricity and the feasibility of techniques to balance the difference between energy demand and electricity generation[1]. Hydrogen has emerged as a promising clean energy vector to decarbonize energy systems and achieve carbon-neutral energy cycling. Specifically, the intermittent renewable electricity generated from wind and solar can be stored in the form of hydrogen via water electrolysis and can be redistributed and used in various applications. Proton exchange membrane (PEM) water electrolysis is a means of producing green hydrogen that is compatible with intermittent electricity input[2], though its development is currently impeded by the limited options of efficient anode catalysts that can drive the oxygen evolution reaction (OER) in acidic and oxidative conditions.

Recently, Co-based oxides[3–9] have attracted great interest as a potential replacement for Ir-/Ru-based materials[10,11] as OER catalysts in acidic environments. Specifically, $Co_3O_4$ spinel oxides have been thoroughly investigated, and often act as a platform for heteroatom doping[3,7,8] or in the construction of composites[4] to enhance their OER performance. Moreover, in-situ characterizations[4,5,7] reveal that $Co_3O_4$ spinel oxides undergo a surface reconstruction into CoOOH-like structures under OER conditions in acidic electrolytes, similar to that

[1]PSI Center for Energy and Environmental Sciences, PSI, Villigen, Switzerland. [2]Department of Energy Conversion and Storage, Technical University of Denmark, Kongens Lyngby, Denmark. [3]PSI Center for Photon Science, PSI, Villigen, Switzerland. [4]PSI Center for Life Sciences, PSI, Villigen, Switzerland. [5]Institute for Molecular Physical Sciences, ETH Zürich, Zürich, Switzerland. ✉e-mail: jinzhen.huang@psi.ch; emiliana.fabbri@psi.ch

in alkaline electrolytes[12,13]. The Co[III]OOH-like structures can further evolve into Co[IV] species with increasing applied potential[14,15]. However, not all spinel oxides can drive the OER in acidic electrolytes. Previously, we showed that the surface reconstruction and OER activity of Co oxides in acidic environments are dependent on the oxidation/spin state of Co at the surface[6]. Since Co[II/III] oxidation is not favorable on a pure Co[II] surface, the presence of Co[III] species is important to promote the formation of the active phase, unlike in alkaline electrolytes[13]. Recently, Ram et al. [5] showed that a chemical delamination treatment via anion exchange on $CoWO_4$ (i.e., immersing it in 0.1 M KOH before use) can increase the average Co oxidation state to be close to 2.67+ in $Co_3O_4$, boosting both the activity and stability in acidic electrolytes. More importantly, they also observed reconstruction of delaminated $CoWO_4$ into CoOOH by in-situ Raman characterization, including the formation of Co[IV] species under OER conditions. In general, we conclude that surface reconstruction of cobalt oxides into a CoOOH-like structure is an extremely important factor in the OER performance in acidic electrolytes. Then, one can immediately anticipate that a direct CoOOH structure should be able to drive the OER in acidic environments, but this has not yet been explored. CoOOH is a very well-known OER catalyst in alkaline and neutral environments[14,16], with a structure closer to the final, established OER active phase than $Co_3O_4$. Therefore, it is a better platform to investigate the fundamental electrochemical properties of Co-based catalysts. Specifically, due to changes in the interfacial water structure as a function of electrolyte pH[17,18], a systematic investigation of the electrochemical behavior of CoOOH in acidic electrolytes can help to advance the understanding of the interfacial interactions occurring, thus promoting the design of more efficient Co-based catalysts. However, characterizing the relative stability of CoOOH in an acidic environment is a prerequisite for such an investigation. Interestingly, we find that CoOOH shows comparable stability to $Co_3O_4$ in an acidic environment. Motivated by this, a detailed electrochemical analysis combined with operando hard X-ray absorption spectroscopy (hXAS) at the Co-K edge and density functional theory (DFT) simulations was performed to compare the interfacial properties of CoOOH in acidic environments with those in alkaline environments.

Furthermore, we bridge the gap between the catalyst performance measured in a conventional three-electrode setup and in a single PEM water electrolyzer, a connection of particular relevance here, given the limited understanding of the behavior of Co-based catalysts at the device level. Employing CoOOH as the anode catalyst, the electrochemical performance was compared with other Co-based oxides and the $IrO_2/TiO_2$ benchmark. A stable cell potential was achieved for 400 h at 100 mA cm$^{-2}$ without noticeable degradation. Therefore, the unexplored CoOOH catalyst demonstrates promising performance for its application in PEM water electrolysis. Further performance improvements are anticipated if the open issues of the onset potential for the OER, electric conductivity, and Co dissolution at high current densities are addressed.

## Results

### Structure characterizations of CoOOH

The CoOOH catalyst was synthesized by a two-step wet-chemistry process (see the "Methods" for details). The crystal structure was confirmed by powder X-ray diffraction (XRD) (Fig. 1a). The analysis of the XRD pattern reveals that the as-prepared CoOOH is a typical 3R polytype structure[19] with lattice parameters of $a = b = 2.85$ Å, and $c = 13.18$ Å (Supplementary Table 1). There are no intercalated ions between the $CoO_6$ octahedral layers, different from the non-crystalline layer Co oxide catalysts[20,21]. Transmission electron microscopy (TEM) characterizations demonstrate the polycrystalline nature of CoOOH (Supplementary Fig. 1). Under high-resolution TEM, the alignment of atoms in CoOOH is quasi-hexagonal, with a measured distance of ~2.6 Å (Fig. 1b). The distance between two atom arrays is ~2.3 Å,

matching the distance of (012) planes in CoOOH. Ex-situ hXAS characterization at the Co K edge was performed to uncover the Co oxidation state and coordination environments of CoOOH. Commercial $Co(OH)_2$, $Co_3O_4$, and $LiCoO_2$ with a theoretical average Co oxidation state of 2+, 2.67+, and 3+, respectively, are used as reference samples. The X-ray absorption near-edge structure (XANES) spectrum of the as-prepared CoOOH is close to that of commercial $LiCoO_2$ at the Co K edge energy region, consistent with an average Co oxidation state of 3+ for CoOOH (Fig. 1c). Due to inequivalent chemical environments in the two samples, the edge features are different. Therefore, the energy of the Co K edge ($E_{edge}$) extracted using an integral method[22,23] is 7722.10 eV for CoOOH, higher than the 7721.83 eV observed for commercial $LiCoO_2$ (inset in Fig. 1c). The corresponding Fourier transformed extended X-ray absorption fine structure (EXAFS) spectrum of CoOOH shows two distinct peaks, corresponding to the Co-O and Co-Co scattering paths, respectively (Fig. 1d). The Co-O and Co-Co distances extracted from fitting are ~1.90 Å and 2.85 Å, respectively (Supplementary Table 2). In summary, the targeted CoOOH catalyst has been successfully synthesized and will be further analyzed by operando hXAS and electrochemical characterizations below.

### pH-dependent Co oxidation transformation

CoOOH is a well-known catalyst that shows good OER performance in alkaline and neutral environments[14,16], but it has not been well studied in acidic environments. Therefore, cyclic voltammograms (CVs) were recorded in both acidic and alkaline electrolytes (pH ~1 and 13, respectively), in addition to two other buffered electrolytes of pH ~7 and 10, to better capture the effect of pH on the redox characteristics of CoOOH. The CVs shown in Fig. 2a indicate that the Co redox features are notably altered by the pH environment (see also Supplementary Fig. 2). In an alkaline electrolyte (pH ~13), two pairs of redox peaks are assigned to Co[II/III] and Co[III/IV] redox processes at the low and high applied potentials, respectively[22]. In a neutral electrolyte (pH ~7), the Co[III/IV] reduction peak is still identifiable, but the Co[III/IV] oxidation peak is not. Furthermore, there is only one discernible redox pair observed in the acidic electrolyte (pH ~1). These CVs recorded at different pHs show a clear super-Nernstian shift in the Co redox peaks, as the numbers of electrons and protons transferred during the Co redox processes are not equal, which is commonly found in Co-based catalysts[23,24]. In addition, a similar super-Nernstian shift in the redox peaks is also evidenced in a $Co_3O_4$ control sample (Supplementary Figs. 3 and 4), as its surface will evolve into CoOOH under OER conditions. For CoOOH, the super-Nernstian shift is more notable in the Co[II/III] redox peaks, causing them to overlap with the Co[III/IV] redox peaks in an acidic environment. Henceforth, the single pair of redox peaks observed in an acidic electrolyte is labeled as Co[II/III] redox, though a contribution from the Co[III/IV] redox process cannot be excluded. The Co[II/III] redox formal potentials ($V^o(Co^{II/III}) = (V_a + V_c)/2$, where $V_a$ and $V_c$ are the potentials of the anodic and cathodic redox peaks, respectively) were extracted from the CVs at different scan rates (Supplementary Fig. 2), as summarized in Fig. 2b. The pH dependence of $V^o(Co^{II/III})$ is clearly evident. To confirm that the observed electrochemical behavior originates from the CoOOH structure and not from specific sample characteristics, another CoOOH catalyst with higher crystallinity was prepared by oxidizing commercial beta-phase $Co(OH)_2$ (denoted as b-CoOOH, Supplementary Fig. 5). Likewise, the as-prepared b-CoOOH also only shows one redox pair in an acidic environment (Supplementary Fig. 6).

To further elucidate the oxidative Co transformation in CoOOH, operando hXAS characterizations[25,26] were also performed in both acidic and alkaline electrolytes using a spectro-electrochemical flow cell[27] (Supplementary Fig. 7, see "Methods" for details). A chronopotentiometry (CP) protocol comprising 21 steps (30 s per step) was applied to the CoOOH catalyst. The constant current was increased from 0.001 mA in the 1st step to 1 mA in the 21st step, and the applied

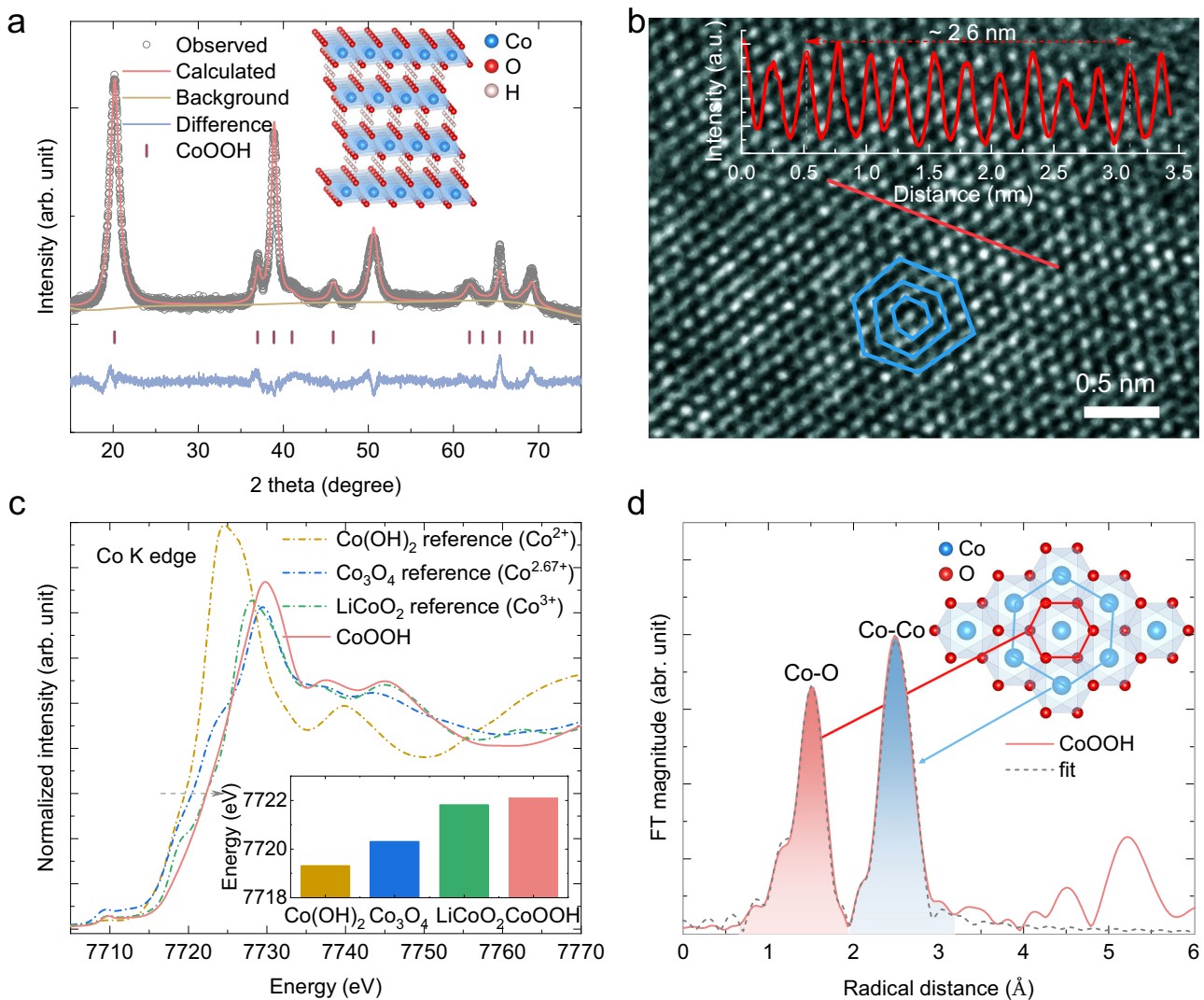

**Fig. 1 | Structural characterizations of CoOOH. a** Rietveld refinement of the XRD pattern. The observed data are plotted in open mark. The calculated pattern (red curve), the background (yellow curve), and the difference between the observed data and calculated pattern (blue curve) are also demonstrated. The standard pattern for CoOOH is shown for reference. The inset shows the layered structure of CoOOH (Co atoms in blue, O atoms in red, H atoms in pink). **b** High resolution TEM images. The inset shows the intensity profile of the red straight line. The blue hexagonal shape shows the atomic alignment. **c** XANES spectra at the Co K edge of commercial $Co(OH)_2$, $Co_3O_4$ and $LiCoO_2$ reference samples, with an oxidation state of $Co^{2+}$, $Co^{2.67+}$ and $Co^{3+}$, respectively. The inset shows the energy of the Co K edge ($E_{edge}$) extracted by an integral method. **d** Fitting of the $k^3$-weighted Fourier transformed EXAFS spectrum at the Co K edge of CoOOH, with the first and second shell of shaded in red and blue, respectively. The inset shows the layered atomic structure of CoOOH. Source data are provided as a Source Data file.

potentials (Supplementary Fig. 8) and the hXAS spectra at the Co-K edge (Fig. 2c, d and Supplementary Fig. 9) were recorded in 0.05 M $H_2SO_4$ and 0.1 M KOH, respectively. The edge position of the Co-K edge spectra ($E_{edge}$) shifts to higher energies as the applied current increases, indicating an increase in the average Co oxidation state (Fig. 2c). To semi-quantitatively present the changes in the average Co oxidation state, the $E_{edge}$ was first extracted by an integral method[26,28], and then the change in the $E_{edge}$ at each step (i.e., $\Delta E_{edge}$) was calculated by subtracting the $E_{edge}$ in the 1st step. We note that the $\Delta E_{edge}$ determined by the integral method is equivalent to the conventional method of using the energy at half the edge-jump intensity (Supplementary Fig. 10). The $\Delta E_{edge}$ is plotted as a function of the applied potential and compared to the corresponding redox features in acidic and alkaline electrolytes (Fig. 2e, f). Co starts to be oxidized at ~1.20 V vs. RHE in an alkaline electrolyte, close to the corresponding $V^o(Co^{II/III})$ as determined from the CV measurement in Fig. 2a, b. In an acidic electrolyte, the start of Co oxidation, indicated by an increase in $\Delta E_{edge}$, is postponed to ~1.40 V vs RHE, before the corresponding

$V^o(Co^{II/III})$ of 1.58 V vs RHE. By comparing $\Delta E_{edge}$ with the CVs, we uncover that the observed pH-dependent redox features are coupled with the differences in the interfacial Co oxidative transformations as a function of the electrolyte pH. In addition, the onsets of $\Delta E_{edge}$ (i.e., ~1.20 and ~1.40 V vs. RHE in pH = 1 and 13, respectively) are also located at around the flat band potential of CoOOH ($1.15 \pm 0.05$ and $1.51 \pm 0.07$ V vs. RHE in pH = 1 and 13, respectively, Supplementary Figs. 11 and 12), which is consistently observed for Co-based catalysts[24,26,29]. Due to the bulk averaging characteristics of the hXAS technique, there is no notable change in the operando EXAFS spectra (Supplementary Fig. 13). Both the bond length and coordination number from fitting remain similar at different currents (Supplementary Fig. 14), suggesting that only a thin surface layer of CoOOH is reconstructed and oxidized to $Co^{IV}$ species[15] (or $Co^{3+\delta}$ referred to ref. 30) under OER conditions.

Surface-sensitive soft XAS characterizations at the $Co\ L_{2,3}$ edge and O K edge were performed to uncover the surface changes in CoOOH before and after the OER measurements (Fig. 2g, h). CoOOH

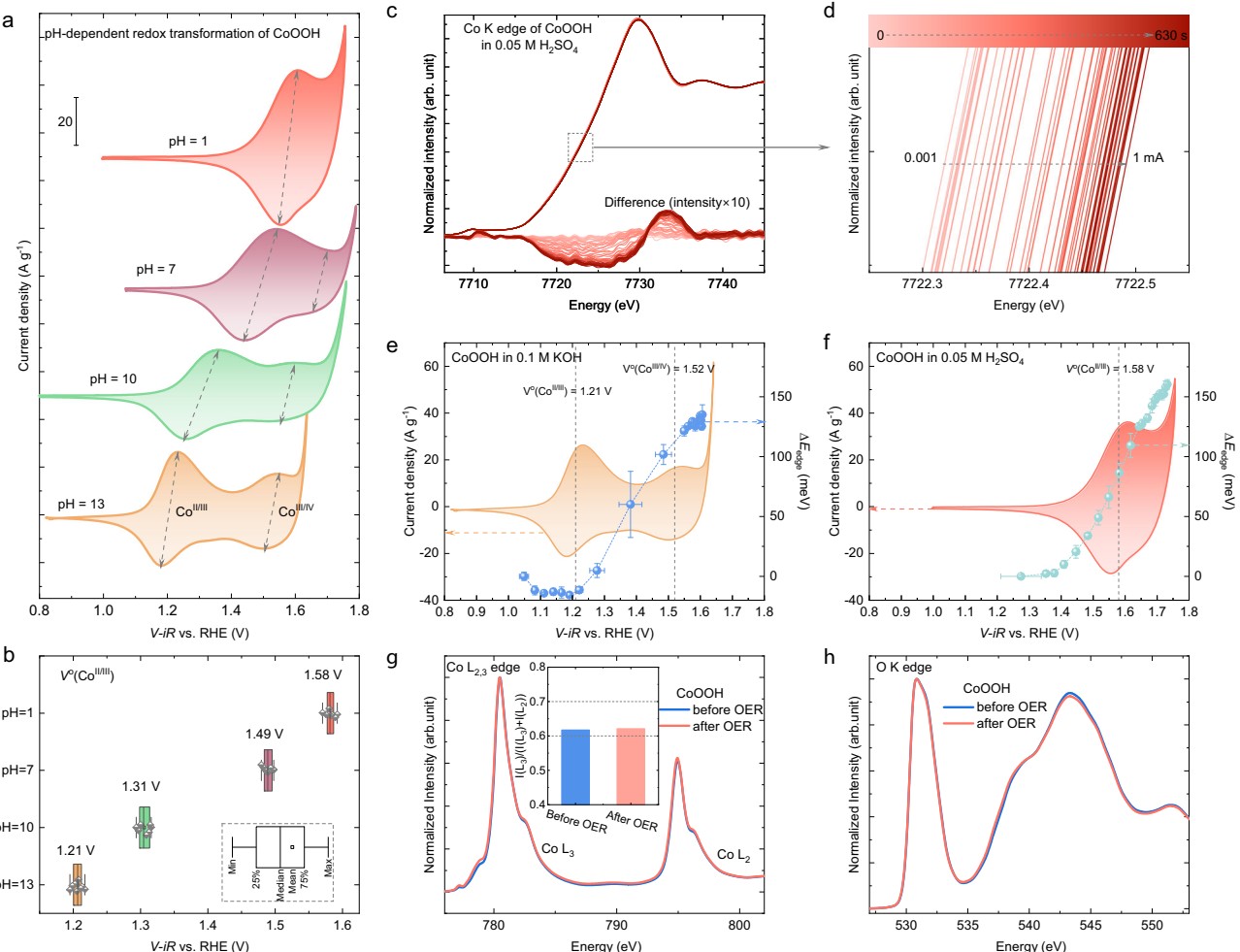

**Fig. 2 | The pH-dependent Co redox process and oxidative transformations of CoOOH.** The pH-dependent Co redox transformation indicated by **a** CVs collected at the scan rate of 100 mV s$^{-1}$ and in electrolytes of pH = -13, 10, 7 and 1, and the resistances for *iR*-correction are 54 ± 1.6, 104 ± 9, 85.4 ± 0.2 and 40 ± 1 Ω, respectively, with standard error from three replicates. The catalyst loading is ~20 μg. **b** The corresponding formal redox potential of Co$^{II/III}$ (denoted as $V^o$(Co$^{II/III}$)). **c–f** Operando hXAS characterizations during an electrocatalytic CP protocol whereby a constant current is held for 30 s at each step. The operando hXAS spectra at the Co-K edge collected in 0.05 M H$_2$SO$_4$ are shown in (**c**, **d**). The energy shift of the Co K edge ($\Delta E_{edge}$) is plotted as a function of the applied potential in **e** 0.1 M KOH and **f** 0.05 M H$_2$SO$_4$, compared to the redox process shown by the CVs. The standard error for $\Delta E_{edge}$ is obtained by averaging three normalized spectra collected in a 30 s period. The standard error for the applied potential results from averaging the measured potentials over the 30 s period. Surface-sensitive soft-XAS measurements at the **g** Co L$_{2,3}$ edge and **h** O K edge of CoOOH before and after OER in an acidic environment. Source data are provided as a Source data file.

shows the strongest peak intensity at ~780 eV at the Co L$_3$ edge, indicating that the surface is dominated by Co$^{III}$ species[26,29]. The branching ratio of I(L$_3$)/(I(L$_3$) + I(L$_2$)), calculated from the intensity of the Co-L$_2$ and Co-L$_3$ edges, is ~0.62 (inset in Fig. 2g), close to the theoretical value of 0.6 for Co in a low-spin state[6,31]. In addition, the strong pre-edge peak before 535 eV in the O K edge spectrum supports the existence of low-spin Co$^{III}$ (see ref. [32]). Therefore, the surface of CoOOH is dominated by the low-spin Co$^{III}$ species and is active toward the OER in acidic electrolytes, in line with the findings from our previous work[6]. More importantly, there is no change in both the Co-L$_{2,3}$ edge and the O-K edge after OER tests in acidic electrolytes. Therefore, though the low-spin Co$^{III}$ species at the surface of CoOOH can be further oxidized into Co$^{IV}$ species (or Co$^{3+\delta}$ referred to ref. [30]) under OER conditions, they reversibly return to the initial state when CoOOH is taken out of the electrolyte and dried.

## Redox charge at different interfaces

The redox charge passed during the interfacial oxidation and reduction of Co species can be estimated by calculating the total charge from the area of CVs (see Supplementary Fig. 2 for more details). Due to the overlap of the OER with the Co redox processes and also the contributions from double-layer capacitance, the as-calculated redox charge is slightly overestimated. To reduce these influences, the redox charges are extracted from CVs collected at different scan rates, and their values are summarized in Fig. 3a. Although the formal redox potential shifts as a function of the electrolyte pH, the estimated average redox charges extracted in different electrolytes remain quite constant at around 140–150 C g$^{-1}$, suggesting that the coverage of redox-active species in CoOOH is only slightly affected by the pH environment. In comparison, the pH-induced change in redox charges is more pronounced in the Co$_3$O$_4$ control sample (Supplementary Fig. 4), which could be explained by the unfavorable Co$^{II}$ to Co$^{III}$ oxidation that occurs in acidic environments, but not in alkaline environments[6,24]. Furthermore, charge accumulation at the interface plays an important role in governing the OER activity[33,34], and usually results in a linear correlation in the log-log plots of OER current vs. total charge. A similar correlation can be drawn by plotting the measured potential and the $\Delta E_{edge}$ of the Co K edge in different electrolytes (Fig. 3b, c), showing the charge accumulation via oxidation of Co centers as a function of the logarithm of current (i.e., log(*i*)).

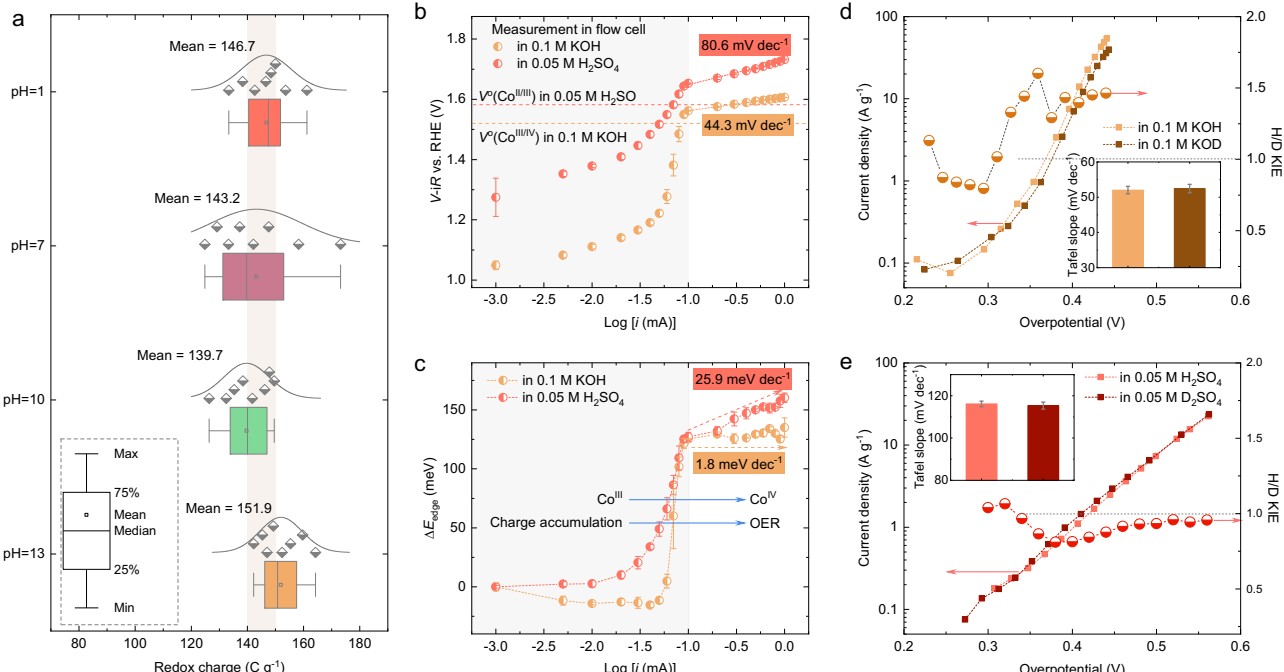

**Fig. 3 | Redox charge and proton transfer in the OER. a** Redox charges (the scatters) estimated by the area of CVs collected at different scan rates and in different electrolytes in Supplementary Fig. 2. The distribution of the redox charges is indicated by the curve above the scatters. **b** Tafel plots derived from the CP measurement during operando hXAS characterizations in alkaline and acidic environments. The standard error for the applied potential results from the average of the measured potential during every 30 s period. The horizontal dashed lines are the corresponding formal redox potentials. **c** The energy shift of the Co K edge ($\Delta E_{edge}$) is plotted as a function of log[$i$(mA)]. The standard error bars are from the average of three spectra collected during every 30 s period. Deuterium substitution in the electrolyte to study the H/D KIE in **d** 0.1 M KOH/KOD and **e** 0.05 M H₂SO₄/D₂SO₄. The insets in (**d**, **e**) are the corresponding Tafel slopes with error bars from fitting. Source data are provided as a Source data file.

From the CP measurements performed in the spectro-electrochemical flow cell (Supplementary Figs. 7 and 8), plots of the applied potential as a function of log($i$) for CoOOH in acidic and alkaline electrolytes were obtained and are shown in Fig. 3b. This plot can be divided into two regions for both electrolytes: in the low current region (log($i$) < −1), the applied current in the CP measurement is first used to charge the CoOOH and drive the surface reconstruction, with faster increase in the measured potential occurring closer to the formal redox potential (horizontal dashed lines in Fig. 3b); when the measured potential is above formal redox potential and log($i$) ≥ −1, the measured potential is attributed to the OER at the catalyst surface, and it displays a typical linear correlation of applied potential against log($i$) as seen in the Tafel plot. The Tafel slopes from linear fitting in alkaline and acidic electrolytes are 44.3 mV dec⁻¹ and 80.6 mV dec⁻¹, respectively, indicating faster reaction kinetics are occurring in the alkaline conditions. Interestingly, when the $\Delta E_{edge}$ of the Co K edge is also plotted as function of log($i$) in Fig. 3c, a similar trend to the applied potential vs log($i$) relation is observed. When applied current is used to charge the CoOOH to drive surface reconstruction, the $\Delta E_{edge}$ first increases slowly and then much more notably when close to log($i$) = −1; in addition, the increase of $\Delta E_{edge}$ at log($i$) = −1 is almost the same in both alkaline and acidic electrolytes (-125 meV). In the OER region (log($i$) ≥ −1), the $\Delta E_{edge}$ measured in an alkaline electrolyte almost does not change as a function of the control current, with a very small slope of 1.8 meV dec⁻¹ observed, in stark contrast to a higher slope of 25.9 meV dec⁻¹ in an acidic electrolyte. To summarize, the present study shows that both the redox charges extracted by CV measurement (in Fig. 3a) and the extent of the changes in the Co oxidation at log($i$) < −1 (see $\Delta E_{edge}$ in Fig. 3c) are almost the same in alkaline and acidic electrolytes. These results differ from those found in a commercial CoO$_x$ (ref. 24), in which the redox charge and the extent of changes in the Co oxidation transformation are pH-dependent. This is

likely due to the fact that Co$^{II/III}$ oxidation is less favorable in an acidic electrolyte than in an alkaline electrolyte[6]. These findings suggest that the surface reconstruction to form the active sites is still occurring for CoOOH both in acidic and alkaline environments, with the redox charge related to the abundance of redox-active Co species at the interfaces unaffected by the pH environment. Therefore, it is not the main factor responsible for the different activity in acidic vs. alkaline electrolytes for the CoOOH catalyst.

The different activity in acidic or alkaline electrolytes could originate from different reaction intermediate interactions. For example, it has been found that the reaction intermediates formed in IrO$_x$ under OER conditions exhibit repulsive lateral interactions among each other[17,35]; and the binding energies of the reaction intermediates and their repulsive interactions are weaker in acidic electrolytes compared to alkaline electrolytes. Then, the change in Tafel slopes of CoOOH between acidic and alkaline electrolytes could be related to the differences in the potential-dependent coverage of reaction intermediates[17]. Moreover, an anodic shift of the formal redox potential could also be explained as a decrease in the binding energy of oxygen intermediates[36,37]. Here, the notable anodic shift of $V°(Co^{II/III})$ for CoOOH in low pH environment (in Fig. 1a, b) supports a weaker binding of reaction intermediates in acidic vs. alkaline electrolytes. Therefore, the smaller Tafel slope in an alkaline electrolyte could be due to the stronger binding energies of intermediates compared to those in an acidic electrolyte[17]. Furthermore, Pasquini et al.[28] proposed that the interactions between Co centers in Co-based oxides can hinder the further oxidation of a neighboring Co atom near the oxidized ones, resulting in a higher potential needed to drive Co oxidation in the OER region. In this study, a weaker repulsive intermediate interaction in CoOOH results in a higher slope of 25.9 meV dec⁻¹ in the plot of $\Delta E_{edge}$ vs log($i$) in an acidic electrolyte (Fig. 3c), suggesting that increasing the OER rate is more dependent on the Co oxidation state at

this interface. Therefore, the present results suggest that it is not the pH-dependent intermediate surface coverage, but the intermediate interactions that play an important role in determining the pH-dependent OER kinetics of CoOOH. We acknowledge that other parameters, e.g., water dissociation or interfacial water reorganization etc.[38] might also contribute to the change in Tafel slope in different pH environments, and need to be further verified in future studies. Since the change in Co oxidation state at the different CoOOH/electrolyte interfaces reflects the synergic contributions of these parameters, they directly connect the interfacial charge to the OER current[33]. The present results suggest that it is not the pH-dependent surface coverage of intermediates formed before the onset of OER, but the intermediate interactions that play an important role in determining the Co oxidation under OER conditions, resulting in the pH-dependent OER activity of CoOOH.

## Proton transfer during OER

The origin of kinetic isotope effect (KIE) theory is that changing proton (H) to deuterium (D) can modify the zero-point energies of O−H and O−D bonds[39]. In the conventional adsorbate evolution mechanism (AEM), proton and electron transfer are concerted during the OER, and observation of a primary KIE is expected (i.e., KIE ≥ 2) when a H is substituted by D in the electrolyte[28]. To study the H/D KIE for CoOOH in both alkaline and acidic electrolytes, new electrolytes of 0.1 M KOD (purified by $Co(OH)_2$) and 0.05 M $D_2SO_4$ were prepared (See "Methods" for the details). The H/D KIE is extracted from the ratio of the current density (KIE = $I_H/I_D$) at the same overpotential ($\eta$), with the procedure discussed in "Methods" and Supplementary Figs. 15 and 16. The polarization curves (with the logarithm of current density) in both H- and D-electrolytes and the corresponding KIE as a function of overpotential are shown in Fig. 3d, e. A similar OER performance (on the overpotential scale) can be observed after D substitution in both acidic and alkaline electrolytes. The calculated KIE is around 1, signifying a secondary KIE and indicating that protonation/deprotonation is not involved in the rate-determining step (RDS) in both alkaline and acidic electrolytes[28]. In addition, the Tafel slopes (insets of Fig. 3d, e) derived from the polarization curves also demonstrate no notable changes, further proving that the reaction kinetics are not changed by H/D substitution in both alkaline and acidic electrolytes.

It should be noted that the OER reactants are different in alkaline electrolytes ($OH^-$) compared to acidic electrolytes ($H_2O$). If the CoOOH follows a conventional AEM in alkaline electrolytes with *O + $OH^-$ → *OOH + $e^-$ (* represents the active site) as the RDS[40], then it is reasonable to observe secondary KIE for OER, since under certain conditions there is no break of O−H bond (details outlined in Supplementary Note 1 and Supplementary Table 3). In comparison, when $H_2O$ is the reactant in acidic electrolytes, a break of an O-H bond happens at every step during water dissociation or deprotonation, including in the formation of *OOH (*O + $H_2O$ → *OOH + $H^+$ + $e^-$). Since proton and electron transfer are concerted in the AEM pathway[28], then a primary KIE (KIE ≥ 2) and a change in the Tafel slope would be expected; however, this does not match with the observation in Fig. 3e. Therefore, the absence of a primary KIE suggests that proton transfer is decoupled from the electron transfer in the RDS of a CoOOH catalyst in an acidic electrolyte.

## OER mechanism

To obtain more insights on the pH-dependent OER activity of CoOOH, we collected Tafel plots in a temperature range of 30–70 °C in different electrolytes (Supplementary Fig. 17) to extract the apparent activation energy ($E_{app}$) and the logarithm of the apparent exponential factor log($A_{app}$), according to the Arrhenius equation [ $j = A_{app}$*exp($-E_{app}/RT$), details in "Methods"][4,41]. The Tafel plots of CoOOH collected at 30 °C in different electrolytes are shown in Fig. 4a. We defined the applied potential vs. RHE at the current density of 1 A $g^{-1}$,

i.e., log[$j$ (A $g^{-1}$)] = 0, as the onset potential for the OER in different electrolytes (inset in Fig. 4a). Unlike the $V^o(Co^{II/III})$, which increases monotonically when reducing the pH of the electrolyte (in Fig. 1b), the OER onset potential increases in the order: alkaline (-1.58 V) <acidic (-1.62 V) <neutral (-1.67 V). The inconsistency between the $V^o(Co^{II/III})$ and onset potential could be due to the synergic influence of several factors, including electro-adsorption energy[18], interfacial water structure[17], active-site coverage[35], and so on. A similar observation has been reported for the redox potentials in $RuO_2$ (ref. 18), which are associated with the binding energies of OER intermediates and anodically shift in lower pHs, while the OER activity displays a different trend, with the largest overpotentials seen in a neutral electrolyte.

Note that the Arrhenius equation shows the relationship between the OER rate, which correlates with OER current density (j), and the temperature[41,42]. The $E_{app}$ and log($A_{app}$) can be experimentally determined from the Arrhenius plot (log(j) vs. 1000/$T$). However, the non-Nernstian overpotential shift induced by the electrolyte pH suggests that the formation of charged reaction intermediates is pH-dependent[38]. In addition, the extracted $E_{app}$ and log($A_{app}$) are linearly scaled and are potential-dependent[4,41], and the entropic changes in different interfacial solvent has notable influence on the extracted log($A_{app}$)[38,42]. The pH-induced shift of the OER onset potential in different electrolytes prohibits a fair comparison of the $E_{app}$ at the same overpotential in different electrolytes, as the oxidation of Co to form reaction intermediates is delayed in lower pH environments (as shown in Fig. 2e, f). Instead of using a constant overpotential to compare the $E_{app}$ for different electrolytes, a comparison at the OER onset potentials (inset in Fig. 4a) for each electrolyte can provide a comprehensive understanding of its pH dependence, as the OER is controlled by the reaction kinetics. Accordingly, log(j) is first plotted as a function of 1000/$T$ to extract the $E_{app}$ and log($A_{app}$) in different electrolytes (Fig. 4b and Supplementary Table 4). Finally, log($A_{app}$) is plotted as a function of $E_{app}$ to derive a strong linear relationship (inset in Fig. 4b), arising due to the compensation effect of the entropy of activation (related to $A_{app}$) with the enthalpy of activation (related to $E_{app}$)[42,43]. The extracted $E_{app}$ for CoOOH increases with the pH of the electrolyte, indicating that the pH of the electrolyte influences the activation energy of the formation of OER intermediates.

Generally, the conventional AEM is not adequate to explain the absence of a primary KIE, the pH dependence of the OER activity, and the apparent activation energy $E_{app}$. Therefore, DFT calculations were performed using the (10−14) surface of CoOOH as a model surface (more details in "Methods"), which has been reported to be the most stable, with a coverage of 1 mL $H_2O$ which dissociates immediately into *OH (ref. 44). Here, the highest energy barrier is found for the formation of *OOH in the AEM pathway, which is pH-independent (Supplementary Fig. 18), implying that an alternative reaction mechanism should be considered to better explain the experimental observations. Recently, the OPM has been frequently reported for OER catalysts[5,7,45]. In particular, the local bonding environments of CoOOH (Supplementary Table 2) are similar to those of a CoPi catalyst consisting of similar $CoO_6$ slabs[46], which meets the geometric requirement for the OPM[47]. A similar mechanism has also been proposed for reconstructed CoOOH in an acidic environment[5]. Therefore, the OPM was investigated as a potential reaction pathway for the directly synthesized CoOOH (Fig. 4c), featuring the direct coupling of two adjacent *O species, which are axially adsorbed at the surface, instead of *OOH formation as observed in the AEM pathway. The calculated energies for the formation of subsequent intermediates (not including the transition state[45]) along with the reaction coordinates are shown in Fig. 4d, and the effect of both the electrolyte pH (pH = 1, 7, and 13) and potential ($U$ = 1.23 V) are considered and corrected during the calculations (details in "Methods"). The calculated energy for the O-O coupling is higher compared to other steps and is pH-dependent, increasing with the pH of the electrolyte and in line with the

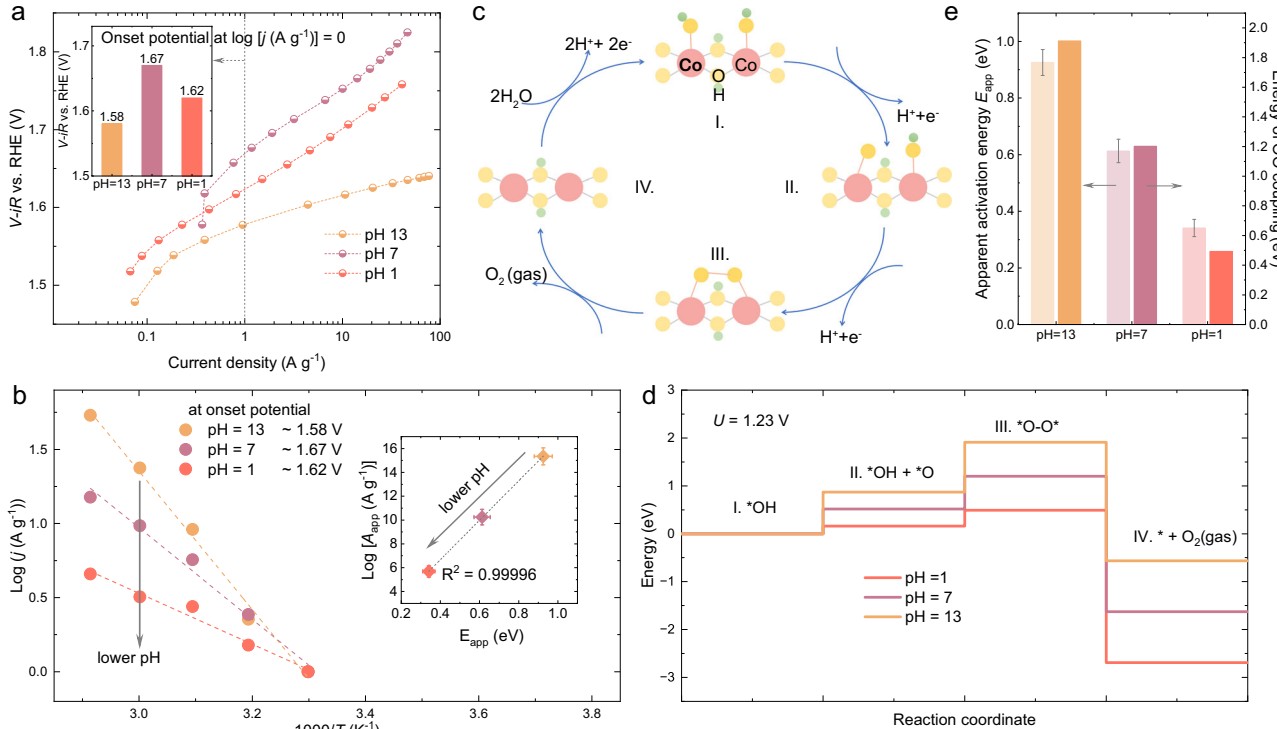

**Fig. 4 | Experimental and theoretical verification of the OER mechanism for CoOOH. a** Comparison of the OER performance of CoOOH in different electrolytes (pH = -13, 7, and 1) at 30 °C. The inset shows the onset potential for OER at log[$j$ (A g$^{-1}$)] = 0. The Tafel plots collected at other temperatures are shown in Supplementary Fig. 17. **b** In each electrolyte, log[$j$ (A g$^{-1}$)] at the corresponding onset potential is plotted against 1000/$T$ to extract the apparent activation energy $E_{app}$ and the pre-exponential factor $A_{app}$, as summarized in Supplementary Table 4.

The inset shows the linear correlation between the $E_{app}$ and log[$A_{app}$ (A g$^{-1}$)]. The error bars were obtained from fitting. **c** Scheme of reaction intermediates in an oxide path mechanism (OPM) pathway. **d** The energy diagram for the OPM pathway in different pH environments. The results were calculated with an electrode potential of $U = 1.23$ V. **e** Comparison of the apparent activation energy $E_{app}$ in (**b**) with the calculated energy for the OO coupling (i.e., intermediate III) in (**d**). Source data are provided as a Source data file.

experimentally extracted $E_{app}$. A side-by-side comparison of $E_{app}$ with the calculated energy for O−O coupling is made in Fig. 4e to show the good agreement of this trend. Specifically, at pH = 1, the $E_{app}$ is ~0.34 eV (extracted at 1.62 V vs. RHE) and the energy of O−O coupling is ~0.49 eV (calculated at the electrode potential of $U = 1.23$ V). This numerical difference can result from the inherent error arising when the solvation and electric-field effects are ignored[18]. In addition, the calculated energies are also potential-dependent (Supplementary Fig. 19), which can be attributed to the difference between the experimental and calculated results.

Besides the solvation and electric-field effects being ignored, there are some other conditions not included in the DFT model here. For example, though the (10−14) surface of CoOOH is considered to be the most stable[44], it is also widely accepted that surface reconstruction from CoOOH to CoO$_2$ happens under OER conditions[14]. The understanding of the local structure for the reconstructed CoO$_2$ is still lacking, hindering the construction of a DFT model that is closer to the real active phase. In addition, since the water structure changes with the pH of the electrolyte[17], the super-Nernstian shift of the Co$^{II/III}$ redox process[23,28] was observed in CoOOH (in Fig. 2a, b). Consequently, the onset potential for the OER is also delayed in acidic electrolytes with respect to alkaline electrolytes, and this cannot be explained by the lower energy barrier calculated for pH = 1. Therefore, we point out that the calculated results here show good agreement with the experimental $E_{app}$ but cannot capture all the factors that contribute to the pH-dependent OER performance. Besides the OPM mechanism[5,7] proposed here, other mechanisms have also been proposed for the CoOOH and Co-based catalysts[14,23,48]. In addition, the AEM pathway can also co-exist with the OPM pathway[5] or other pathways. Indirect probe methods at this stage make it difficult to capture the dynamic and

complex interfacial electrochemical processes at the molecular scale required to determine the reaction pathways. Therefore, from an experimental point of view, further efforts are required to precisely elucidate the complex transformation of interfacial intermediates under OER conditions.

## Stability in acidic electrolytes
The stability of CoOOH is an essential parameter to consider when evaluating its OER catalytic properties in acidic environments. Therefore, operando hXAS was used to monitor the reversibility of Co oxidation at the surface of CoOOH during CV measurements in 0.05 M H$_2$SO$_4$ (Supplementary Figs. 20 and 21). Both the OER current and the simultaneous changes in the Co-K edge (i.e., $\Delta E_{edge}$) are plotted as a function of time. Figure 5a shows that the $\Delta E_{edge}$ periodically changes with the applied potential. At the end of the third CV cycle, a change of 60 meV of the $\Delta E_{edge}$ arises from the formation of thicker activated layers at the surface, which is commonly observed since Co oxidation in the sublayer is coupled with the formation of a surface layer that directly participates in the OER[24]. However, these oxidized layers may reverse back to Co$^{III}$ state when the catalyst is taken out from the electrolyte and dried, as evidenced by the ex-situ soft XAS characterizations (Fig. 2g, h). Furthermore, we recorded CP curves at a constant current of 1 mA (i.e., ~7.9 mA cm$^{-2}$) for 1 h, and the Co K edge of CoOOH is simultaneously recorded to unveil changes in the average Co oxidation state (Supplementary Fig. 22). The applied potential increases by ~20 mV at the end of the 1 h measurement (Fig. 5b); while the corresponding $\Delta E_{edge}$ at the Co K edge increases by ~80 meV due to the continuous surface reconstruction, similar to the observation after CV measurement. During the CP measurement, there is no obvious change in the operando EXAFS spectra (Fig. 5d). The fitting of the

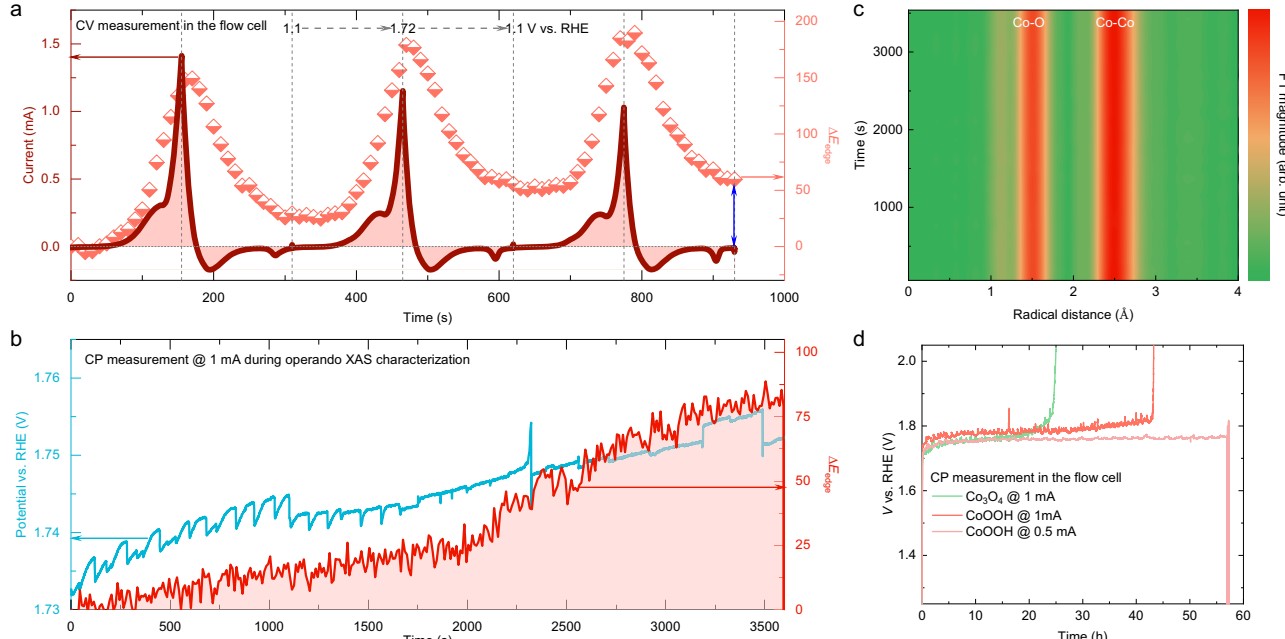

**Fig. 5 | Structural and OER stability of CoOOH in a 0.05 M H₂SO₄ electrolyte.** The energy shift of the Co K edge ($\Delta E_{edge}$) is plotted as a function of time **a** during the CV measurement associated with the reaction current, and **b** during the CP measurement for 1 h associated with the reaction potential. **c** The operando $k^3$-weighted Fourier transformed EXAFS spectra showing the bulk structure of CoOOH during CP measurement for 1 h. **d** The long-term CP curves of CoOOH recorded in the flow cell. The prepared Co₃O₄ is applied as a control sample. Source data are provided as a Source data file.

EXAFS spectra reveals the coordination number and bond length remain unchanged (Supplementary Fig. 23), further confirming that the bulk structure of CoOOH is stable under acidic OER conditions.

We have further performed an extended CP measurement at a constant current of 1 mA in a flow cell without simultaneous operando hXAS characterization (Fig. 5c). The flow cell with CoOOH sprayed onto an Au-coated Kapton foil can operate for ~43 h before drastic changes in the applied potential occur. The sprayed CoOOH was still observed on top of the Au layer, and the breakdown of cell performance is due to the delamination of the conductive Au support (Supplementary Fig. 24). By reducing the constant current by half to 0.5 mA, the operation time was prolonged to ~57 h. In addition, the Co₃O₄ control sample shows comparable performance to CoOOH in the stability test in an acidic electrolyte. According to the Pourbaix diagram, Co is less stable in an acidic environment than in neutral or alkaline environments[44]. The dissolution of Co in the acidic electrolyte after the CP measurement was quantified by inductively coupled plasma optical emission spectroscopy (ICP-OES, Supplementary Table 5). It shows that the Co dissolution in CoOOH is also comparable to that of the Co₃O₄ control sample. These observations convincingly prove that CoOOH can drive the OER in an acidic environment with meta-stability.

## PEM water electrolysis

We further assessed the performance of CoOOH in a PEM water electrolyzer (Fig. 6a) using a home-built electrolysis testbench[49]. A commercial Nafion 212 membrane was used to prepare the catalyst-coated membrane (CCM) for cell measurements (Fig. 6b). The as-prepared CoOOH was employed as the anode catalyst, with an active area of 5 × 5 cm², while the commercial Pt/C was spray-coated onto the cathode side of the CCM (details in "Methods"). The performance of the Co₃O₄ control sample, commercial CoOₓ (from Sigma–Aldrich, Supplementary Fig. 25) and commercial IrO₂/TiO₂ (from Umicore) were also evaluated for comparison. The assembled PEM water electrolysis cell was conditioned at 80 °C by cycling the cell potential between 1.70 and 1.80 V every 300 s for ~45 cycles. During this period, an activation

process was observed (Supplementary Fig. 26), whereby the cell current density gradually increased until stabilization. Electrochemical performance of three individual CCMs with CoOOH as the anode catalyst was evaluated (Supplementary Fig. 27), and the averaged polarization curve is illustrated in Fig. 6c. At a current density of 2 A cm⁻², the cell potential reaches 2.36 V (without $iR$-correction), comparable to that of a La and Mn co-doped Co₃O₄ catalyst (2.47 V in ref. 3) as well as the Co₃O₄ control sample and commercial CoOₓ tested in this study.

Compared to the benchmark IrO₂/TiO₂ catalyst, the cell potential of CoOOH at a low current density of 8 mA cm⁻² is upshifted by ~0.15 V, while at a current density of 2 A cm⁻², a potential gap of ~0.6 V is observed (Supplementary Fig. 28). We attribute the higher cell potential of CoOOH relative to IrO₂/TiO₂ at low current densities (i.e., close to the onset of water electrolysis) to the intrinsic difference in their OER onsets in acidic environments (Supplementary Fig. 29). In addition, the cell high frequency resistance (HFR) measured with CoOOH as the anode catalyst is about three times larger than that with IrO₂/TiO₂ (Supplementary Fig. 30), contributing to a higher ohmic overpotential (~0.41 V in CoOOH compared to ~0.15 V in IrO₂/TiO₂) at a current density of 2 A cm⁻². This higher HFR with Co-based catalysts could be due to their relatively lower electronic conductivity than that of IrO₂/TiO₂, as verified by the ex-situ four-electrode measurement[26,50] (Supplementary Fig. 31). The analysis highlights the importance of ensuring sufficient electronic conductivity in OER catalysts for practical applications[51].

Furthermore, based on an overpotential breakdown analysis of polarization curves, the cell potential can be divided into the reversible potential, kinetic overpotential, ohmic overpotential and rest overpotential (Supplementary Note 2 and Supplementary Figs. 32 and 33). We note that the rest overpotential of CoOOH starts to increase obviously beyond ~500 mA cm⁻², eventually becoming notably higher than that of IrO₂/TiO₂. We expect that the dissolved Co ions at the anode at high current densities may reduce the proton transfer of the Nafion ionomer[52], and can be driven to the cathode by the electric field, causing proton depletion and shifting the reaction mechanism from

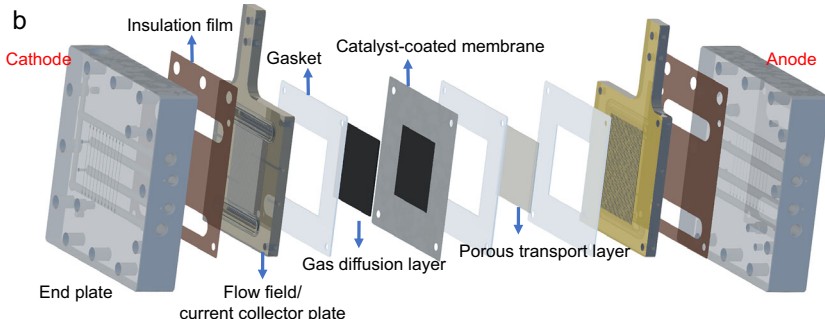

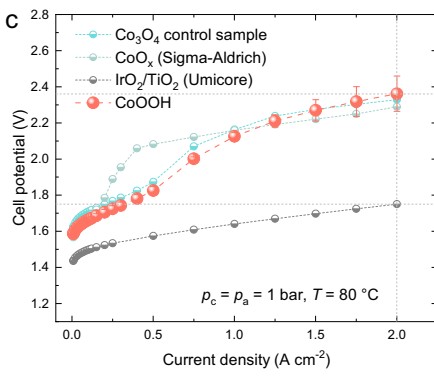

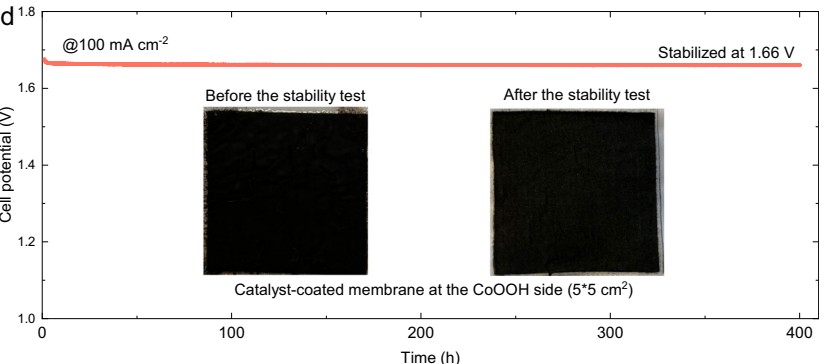

**Fig. 6 | CoOOH in PEM water electrolysis. a** A photograph of the PEM water electrolysis cell used in this study. **b** A schematic representation of the PEM water electrolysis cell, illustrating its various components. **c** Polarization curve of the cell measurement with CoOOH as the anode catalyst, and comparison to those with the $Co_3O_4$ control sample, commercial $CoO_x$, and commercial $IrO_2/TiO_2$. The standard error bar in CoOOH is determined by averaging the results from three individual catalyst-coated membranes. The PEM water electrolyzer was fed with MilliQ water on the anode side and maintained at 80 °C. The gas pressure at both the cathode and anode was maintained at ambient conditions. **d** The CP curve was recorded at a constant current density of 100 mA cm$^{-2}$ for 400 h. The inset pictures show the CCM at the CoOOH side before and after the constant current test. Source data are provided as a Source data file.

proton reduction to water reduction in the cathode[53]. The dissolution of CoOOH is slow at low current densities, as the cell potential at a constant current of 100 mA cm$^{-2}$ is stabilized at ~1.66 V for over 400 h (Fig. 6d). The CCM (insets in Fig. 6d) shows no noticeable changes after the CP test at 100 mA cm$^{-2}$. The measured HFR is found to decrease over time and gradually stabilizes (Supplementary Fig. 34). Specifically, the polarization curve recorded after the CP test shows an overall lower cell potential than that taken before the test, mostly due to a reduced rest overpotential (see Supplementary Note 3 and Supplementary Fig. 35). In contrast, when the applied current density was set to 200 and 500 mA cm$^{-2}$ for the CP measurement, the change in cell potential is evident due to Co dissolution (Supplementary Figs. 36–38).

Overall, CoOOH exhibits relatively stable performance at a lower current density of 100 mA cm$^{-2}$, compared to a La and Mn co-doped $Co_3O_4$ catalyst (200 mA cm$^{-2}$ for 100 h in ref. 3) and a γ-MnO$_2$ catalyst (200 mA cm$^{-2}$ for ~1000 h in ref. 54) in the PEM water electrolyzer. While CoOOH is not stable at higher current densities due to Co dissolution, doping and compositing seem to be promising strategies to address this issue, as proven successfully in the $Co_3O_4$ spinel structure[4,7,8]. Moreover, a water-hydroxide trapping strategy has been applied to increase the stability of a delaminated CoWO$_4$ catalyst[5], enabling a sustained 600 h operation at 1000 mA cm$^{-2}$. The CoOOH catalyst studied here can act as a platform to inspire future research to address the remaining challenges facing the application of Co-based catalysts in PEM water electrolysis.

## Discussion

Although Co-based materials have been studied as OER catalysts in acidic environments, with a view toward application in PEM water electrolysis, understanding of their active phases, interfacial properties and stability is still limited. Here, we demonstrate that the well-known CoOOH catalyst shows promising activity and stability toward the OER in acidic environments. The electrochemical analyses combined with operando hXAS characterizations show that the Co$^{II/III}$ redox of CoOOH is anodically shifted at lower pHs, while the coverage of active sites before the OER onset in acidic electrolytes is comparable to that in alkaline electrolytes. The change in Tafel slopes of CoOOH in acidic vs alkaline environments results from the different potential dependence of Co oxidation under OER conditions, due to the change in interactions between intermediates and interfacial water structure[17]. To align with the experimentally observed pH-dependent trend of apparent activation energy ($E_{app}$) and absence of a primary H/D KIE, an OPM was proposed for CoOOH, which is further verified by DFT calculations. However, other reaction pathways cannot be excluded at this stage. The direct probe methods, which allow operando tracking of the electrochemical surface at the molecular scale and with time resolution, will be beneficial to further clarify the OER pathways in the future.

Additionally, the present study bridges the gap between fundamental understanding and device-level application by employing CoOOH as the anode catalyst in a PEM water electrolyzer. The cell potential stabilizes at ~1.66 V over 400 h of operation at 100 mA cm$^{-2}$, similar to Co- or Mn-based catalysts used for PEM water electrolysis[3,54]. There is still a big gap in the performance of these emerging Earth-abundant catalysts compared to the benchmark Ir-based catalysts in PEM[9]. This study suggests that there are three main parameters that need optimization for the effective use of Co-based catalysts in PEM water electrolysis. First, the OER onset potential of CoOOH is ~0.15 V higher than that of the benchmark $IrO_2/TiO_2$. Then, the intrinsic semiconducting properties of Co-based catalysts result in high HFR under cell operating conditions. Lastly, Co dissolution increases with the current density, leading to performance losses. These findings can

direct further fundamental research on Co-based catalysts towards their successful implementation in PEM water electrolysis, which can inspire future work to realize Co-based catalysts in industrial-scale electrolyzers[55].

## Methods

### Chemicals

$Co(NO_3)_2 \cdot 6H_2O$ (98%), $CoO_x$ control sample (99.99%), $Co(OH)_2$ (99.99%), $Co_3O_4$ reference (99.99%), KOH (99.9%), NaOH (99.9%), $K_2HPO_4$(99%), $KH_2PO_4$(99%), $NaHCO_3$(99%), $Na_2CO_3$ (99%), $Na_2SO_4$ (99%), 2-Propanol and $H_2O_2$ are all from Sigma–Aldrich, Germany. Nafion (99.9%) solution was originally bought from Sigma-Aldrich, Germany. It was mixed with NaOH (99.9%) solution for Na exchange before using electrochemical measurements.

### Synthesis of CoOOH

CoOOH was synthesized by a two-step wet-chemistry protocol[56]. In the first step, 150 mL of 0.1 M NaOH solution was added dropwise into 240 mL of 0.05 M $Co(NO_3)_2$ solution under magnetic stirring at 45 °C. The reaction is kept for 20 min and then the $Co(OH)_2$ precipitate was collected by centrifugation at 4000 rpm. In the second step, the collected $Co(OH)_2$ precipitate was redispersed into 15 mL of 8 M NaOH under magnetic stirring, with the temperature maintained at 45 °C, and then 6 mL of 30% $H_2O_2$ solution was added dropwise slowly. Note that there are massive amounts of oxygen gas generated during the reaction, therefore, the speed of $H_2O_2$ addition should be well controlled. The $Co(OH)_2$ to CoOOH oxidation reaction is kept for 24 h at 45 °C. The CoOOH sample is collected by centrifugation at 4000 rpm (i.e., $1413 \times g$ in relative centrifugal force) and then dried at 60 °C for 24 h.

### Synthesis of b-CoOOH and $Co_3O_4$ control samples

Following a similar method as the second step above, the commercial beta-phase $Co(OH)_2$ from Sigma Aldrich is converted into b-CoOOH, to act as a control sample. In addition, the $Co(OH)_2$ precipitate collected after the first step can also dried at 60 °C for 24 h, and then annealed at 450 °C to obtained the $Co_3O_4$ control sample.

### Physicochemical characterizations

The phase purity and crystal structure of CoOOH were determined by Bruker D8 Advance powder (XRD) at $2\theta = 15-75°$, with a step size of 0.02°. The XRD pattern was refined by the conventional Rietveld method using the General Structure Analysis System−II package[57]. The TEM images were acquired on a Jeol ARM200F (NeoARM) equipped with a cold FEG, probe corrector and OneView camera (Gatan), with a beam energy of 200 keV. The ex-situ soft XAS measurements at the Co $L_{2,3}$ and O K edges were performed at the Xtreme beamline at the Swiss Light Source (SLS), Paul Scherrer Institute (PSI), Switzerland, which utilizes the planar grating monochromator and the optics from the X-Treme beamline[58] using a total electron yield mode. The absorption spectra were obtained as the average of right and left circularly polarized X-ray absorption spectra measured successively at 300 K under 0.05 T applied magnetic field. The CoOOH samples before and after the OER were loaded on a glassy carbon disk for the soft-XAS measurement. The dissolution of Co during the flow cell test was determined by ICP-OES measurement on a 5110 ICP-OES from Agilent, USA.

### Electrochemical characterizations

All electrochemical data were collected using the Biologic VMP-300 potentiostat. For the rotating disk electrode measurement, a home-made Nalgene fluorinated ethylene propylene cell covered with a Teflon cap was used. Four different electrolytes, namely, 0.05 M $H_2SO_4$ (pH = 1 ± 0.05), 0.1 M phosphate buffer (0.054 M/0.064 M of $K_2HPO_4$/ $KH_2PO_4$, pH = 7 ± 0.1), 0.1 M carbonate buffer (0.046 M/0.054 M of $NaHCO_3$/$Na_2CO_3$, pH = 10 ± 0.1), and 0.1 M KOH (pH = 13 ± 0.1) have been used for the measurement. The electrolyte (100 mL) was freshly prepared before each measurement. A $Hg/HgSO_4$ reference electrode (filled with saturated $K_2SO_4$ solution, RE-2CP, ALS, Tokyo, Japan) was used in the electrolytes of pH ~1 and ~7, while a Hg/HgO reference electrode (filled with 0.1 M KOH, RE-61AP, ALS, Tokyo, Japan) was used in the electrolytes of pH ~10 and ~13. The potentials were calibrated to the RHE scale, detailed in Eqs. 2 and 3. A gold mesh was used as the counter electrode in all the electrolytes. The thin-film working electrode[59,60] was prepared by dropcasting 10 μL of 2 g $L^{-1}$ CoOOH ink (i.e., mass loading of 20 μg) onto a glassy carbon disk of 5 mm in diameter (~0.196 $cm^{-2}$). The ink was prepared by dispersing a stoichiometric amount of catalyst in a mixed solution of Milli-Q water (18.2 MΩ·cm), 2-Propanol and Nafion ($Na^+$-exchanged) in a volume ratio of 200:50:1, which was sonicated for 10 min before measurement. The working electrode was rotated at 1600 rpm during the measurement. The OER polarization curves were obtained by performing chronoamperometry for 20 s, and then extracting the average current at each constant potential[26,29] and derive the corresponding Tafel plots. The electrochemical impedance spectroscopy measurement was repeated three times (twice at 1 V vs. RHE and once at 1.6 V vs. RHE), with an amplitude of 5 mV for AC voltage and at the frequency range of 1 M Hz−200 Hz. The average resistance ($R$) is used for $iR$ compensation. The applied potential with iR compensation is denoted as $V$-$iR$, otherwise, it is denoted directly as $V$ in this work.

### H/D kinetic isotope effect (KIE)

Protons in the electrolyte must all be substituted by deuterium. Therefore, $H_2O$ was replaced by $D_2O$ in the preparation of the D-substituted electrolyte. In addition, for the alkaline electrolyte, the KOH was replaced by KOD; while for the acidic electrolyte, the $H_2SO_4$ was replaced by $D_2SO_4$. Note that Fe impurities are reported to notably improve OER performance in alkaline electrolytes. Therefore, to avoid the possible influence of Fe impurities in the D-substituted electrolyte, the alkaline electrolyte used for the H/D KIE study has been purified by the addition of 100 mg of commercial $Co(OH)_2$ into 50 mL of the prepared electrolyte under sonification for 30 min, and then kept overnight. Then the $Co(OH)_2$ was removed from the electrolyte by centrifugation before the measurement. The H/D KIE was calculated by the ratio of current densities at the same overpotential ($\eta$)[4,14]:

$$KIE = \left| \frac{j_H}{j_D} \right|_{\eta} \tag{1}$$

It is challenging to determine the overpotential in the D-substituted electrolytes. First, the potential difference between the reversible hydrogen/deuterium electrode (RHE/RDeE) and the Hg/HgO (or $Hg/HgSO_4$) electrode was determined by recording the CVs in $H_2$-saturated electrolytes (Supplementary Fig. 15). Then, the measured potentials were converted to the RHE/RDeE scale. In acidic conditions, there is no change in the intercept after D/H substitution in the electrolyte, so in both D-electrolyte and H-electrolyte:

$$E(RHE/RDeE) = E(Hg/HgSO_4) + 0.73V \tag{2}$$

In comparison, in alkaline conditions, there is a huge shift in the intercept of CV curves after D/H substitution in the electrolyte, so in H-electrolyte:

$$E(RHE) = E(Hg/HgO) + 0.935V \tag{3}$$

In the D-electrolyte:

$$E(RDeE) = E(Hg/HgO) + 0.976V \tag{4}$$

We note that the equilibrium potential for the OER is shifted from 1.229 V vs. RHE to 1.262 V vs. RDeE after D substitution[61,62], which contributes to the observed anodic shift on the RHE/RDE scale. To obtain the CV curves on an overpotential scale, the conversion in the H-electrolyte is shown in Eq. 5:

$$Overpotenital = E(RHE) - 1.229 V \qquad (5)$$

The conversion in the D-electrolyte should follow Eq. 6:

$$Overpotenital = E(RDeE) - 1.262 V \qquad (6)$$

**Apparent activation energy ($E_{app}$).** The Tafel plots were collected at temperatures of 30, 40, 50, 60, and 70 °C in alkaline, neutral and acidic electrolytes. According to the Arrhenius equation, the OER current density ($j$) can be expressed as[4,41]:

$$j = A_{app} exp(-\frac{E_{app}}{RT}) \qquad (7)$$

where $A_{app}$ refers the apparent pre-exponential factor, $R$ and $T$ are the ideal gas constant and temperature in Kelvin, respectively. Then the $E_{app}$ be extracted at different η by:

$$\frac{E_{app}}{2.303R} = - \left| \frac{\partial \log(j)}{\partial (1/T)} \right|_{\eta} \qquad (8)$$

Log($A_{app}$) is the intercept when log($j$) is plotted as a function of 1/$T$ at each $\eta$.

## Operando XAS characterization in the spectro-electrochemical flow cell

The operando hXAS characterizations were performed at the Super-XAS beamline, SLS, PSI. The incident beam from the 2.9 T super-bend magnetic source was collimated by a Si-coated mirror (2.9 mrad), and monochromatized by a liquid-nitrogen-cooled Si(111) monochromator. A Rh-coated double-focusing mirror was used to obtain a beam size of $1 \times 0.4$ mm$^2$ on the electrode. The ion chambers were filled with 1 bar N$_2$. The transmission signal of a Co foil was measured and used for the energy calibration. The operando experiments were performed in a spectro-electrochemical flow cell, with the cell design detailed in Supplementary Fig. 7 (refs. 27,63). The CoOOH ink (63.72 mg mL$^{-1}$ in a H$_2$O/Nafion mixed solution of v/v = 7/1) was sprayed onto an Au-coated Kapton foil to obtain the working electrode for the flow cell. The sprayed area is ~0.126 cm$^2$, and the mass loading is ~0.12 mg, to achieve a mass loading of ~1 mg cm$^{-2}$. A low-leakage Ag/AgCl reference electrode with a diameter of 2 mm (Harvard Apparatus, USA) was used for the operando measurement. The potential differences of the Ag/AgCl versus Hg/HgSO$_4$ in 0.05 M H$_2$SO$_4$, and versus Hg/HgO in 0.1 M KOH were measured, respectively, to calibrate the measured potential in the flow cell to the RHE scale. The CP protocol consisted of 21 steps with the current increasing from 0.001 to 1 mA. The current was held for 30 s at each step. Two fluorescence spectra at the Co K edge of CoOOH, simultaneously with two transmission spectra at the Co K edge of a Co reference foil, were collected per second. Therefore, in total 60 fluorescence spectra were collected for each step, and every 20 spectra were averaged into 1 spectra during the data calibration and normalization in the ProQEXAFS software[64], to have a time resolution of 10 s. For the CV measurement with a scan rate of 4 mV s$^{-1}$, a resolution of 10 s for the time can also be referred to as a resolution of 40 mV for the applied potential.

## Extraction of $E_{edge}$ of Co K edge

The $E_{edge}$ is determined using the following equation[4,28]:

$$E_{edge} = \frac{1}{\mu_2 - \mu_1} \int_{\mu_1}^{\mu_2} E(\mu) d\mu \qquad (9)$$

where $E(\mu)$ is the energy at the specific normalized absorption intensity, $\mu_1 = 0.2$ and $\mu_2 = 1$ are the lower and upper limit of the normalized intensity. Due to coupling between the operando XAS characterization and the electrochemical measurement in the flow cell, each spectrum collected at a specific time ($t$) can be correlated to the electrochemical parameters, i.e., potential and current ($V$, $i$) of the CoOOH catalyst. Then the $\Delta E_{edge}$ can be calculated to show the energy shift between two different conditions.

## DFT calculation

DFT calculations were performed using the Vienna Ab Initio Simulation Package [65] and the Atomistic Simulation Environment [66]. The exchange-correlation functional used is the revised Perdew-Burke-Ernzerhof [67], with an addition of the Hubbard-U correction[67,68]. The effective Hubbard-U parameter U-J is set to 3.52 eV on the Co-$d$ states[40]. Projector augmented plane wave (PAW) pseudo potentials are used to describe the core electrons[67]. A $3 \times 3 \times 1$ mesh was used for the $k$-point sampling. The cut-off energy is set to 600 eV, with an energy convergence criterion of 10$^{-7}$ eV per cell. Van-der-Walls interactions are included with the Grimme's D3 correction[69]. A dipole moment correction is also included[70]. The mechanism is studied on the (10–14) surface of CoOOH. This has been reported to be most stable with coverage of 1 ML H$_2$O, which dissociates immediately into *OH (ref. 44). The OER can happen through an AEM or an OPM [7]. The correction of pH and potential ($U$) was performed according to:

$$\Delta G = \Delta E - U + k_B T * \ln(10) * pH \qquad (10)$$

where $k_B$ is the Boltzmann constant.

## PEM water electrolyzer measurement

The catalyst-coated membrane (CCM) was prepared by spray coating with an active area of $5 \times 5$ cm$^2$, with Co-based catalysts and Pt/C on the anode side and cathode side, respectively. The anode catalyst ink was prepared by mixing the cobalt (oxyhydr-)oxides powder, Milli-Q water, isopropanol, and Nafion dispersion (Nafion D521CS, 1100 EW at 5 wt% weight percent, Ion Power). The ionomer to (oxyhydr-)oxides weight ratio was 1:9. The cobalt (oxyhydr-)oxides loading was fixed to 2.0 mg/cm$^2$. The Pt on high surface area carbon (TEC10E50E) was used as the cathode catalyst, with an ionomer to carbon ratio of 0.69 and a Pt loading of 0.4 mg/cm$^2$. The prepared catalyst inks were sprayed onto a commercial Nafion 212 (Nafion NR212, IonPower) membrane using an automatic spray coating machine (ExactaCoat, Sono-tek). The prepared CCM was assembled into the electrolysis cell and then mounted onto a home-built electrolysis testbench. All the electrochemical measurements in the PEM water electrolyzer were conducted at 80 °C, and pressure at the cathode and the anode was kept at ambient conditions[71]. For the measurement of each new CCM, the cell was cycled between 1.7 and 1.8 V at an interval of 300 s for at least 4 h until the current density tends to be stabilized at the given voltage level. Electrochemical performance and selected 125-h or 400-h constant current measurements were then carried out.

For the polarization curve analysis, the overpotentials are categorized into three kinds: kinetic overpotential ($\eta_{kinetic}$), ohmic overpotential ($\eta_{ohmic}$) and rest overpotential ($\eta_{rest}$)[49]. Therefore, the cell potential can be described as:

$$V_{cell} = V_{reversible} + \eta_{kinetic} + \eta_{ohmic} + \eta_{rest} \qquad (11)$$

where $V_{reversible}$ is the reversible potential ($V_{reversible}$ = 1.168 V under the conditions of the current study). Specifically, the kinetic overpotential is extrapolated from the fit of the linear part of the Tafel plots.

## Data availability

The source data generated in this study are provided in the Source Data file. The original data for Figs. 1–6 is also available in Materials Cloud Archive, 2025. 105 (2025), https://doi.org/10.24435/materialscloud:ft-jh. Source data are provided with this paper.

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

## Acknowledgements

The synchrotron-based characterizations were carried out at the SLS, PSI, Villigen, Switzerland. E.F. and J.H. gratefully acknowledge the Swiss National Science Foundation through its PRIMA grant (grant No. PR00P2_193111). N.D. gratefully acknowledges the SNFS for financial support under the Ambizione project grant No. PZ00P2_193293.

## Author contributions

J.H. and E.F. develop the concept. J.H. conducted the electrochemical and spectroscopy characterization and analyzed the data. Z.Z. conducted the PEM electrolysis cell, XRD, and ICP measurements. C.S. and I.C. conducted the DFT calculations. A.C., N.H., and N.G. helped to collect and analyze the hard XAS data. N.D. collected the soft XAS data. R.S. collected the TEM images. L.G. helped to analyze the PEM electrolysis cell measurement data. T.S. and E.F. contributed to the results discussion, acquired the funding and provided the research resource. All the authors have revised the approved the manuscript.

## Competing interests

The authors declare no competing interests.
