## [Transparent Peer Review file · Nature Communications]

Directly synthesized cobalt oxyhydroxide as an oxygen evolution catalyst in proton exchange membrane water electrolyzers

Corresponding Author: Dr Emiliana Fabbri

Version 0:

Reviewer comments:

Reviewer #1

(Remarks to the Author)

The authors have conducted a thorough investigation into the pH-dependent OER performance of CoOOH, demonstrating its potential for practical applications in PEM water electrolysis. To explore the underlying factors affecting the OER activity of CoOOH in electrolytes of different pH values, the authors employed electrochemical analysis combined with operando XAS characterizations, providing in-depth insights through data analysis. Furthermore, DFT calculations have verified the oxide path mechanism of CoOOH, which aligns well with the experimental observations. Overall, the manuscript is of high quality in terms of scientific significance, data analysis, and writing style. Therefore, I recommend its acceptance for publication after addressing the following questions.

1. Co₃O₄ typically transforms into CoOOH under OER conditions. Based on this, does Co₃O₄ exhibit a similar pH-dependent OER performance as CoOOH?
2. In Figure 2a, why is only one discernible redox pair observed in acidic electrolyte, whereas two pairs of redox peaks are observed in alkaline and quasi-neutral electrolytes? Additionally, does the redox pair observed in the acidic electrolyte correspond to the CoII/III or CoIII/IV?
3. Under OER conditions, surface reconstruction from CoOOH to the active phase CoO₂ typically occurs. To improve the accuracy of the theoretical calculations, it would be beneficial to use CoO₂ as the model for DFT calculations.
4. Does CoOOH fully follow the OPM pathway, or is it partially following this mechanism? An isotope labeling DEMS experiment could provide additional insights into the OER pathway.
5. After the acidic OER, the Co oxidation state in CoOOH increases (as shown in Figure 5a). The impact of this change on the surface structure of catalyst should be considered.
6. While the oxygen evolution activity of CoOOH shows significant pH dependence, does the stability of CoOOH also exhibit a similar dependence on pH?

Reviewer #2

(Remarks to the Author)

Comments on "Directly synthesized cobalt oxyhydroxide as an oxygen evolution catalyst in proton exchange membrane water electrolyzers"

In this manuscript, the authors demonstrate that the well-known CoOOH catalyst shows promising activity and stability toward the OER in acidic environments. Although the author conducted extensive characterizations and electrochemical analyses to investigate the mechanism of CoOOH, the driving force and innovation of this research appear to be somewhat lacking. This article is still far from being ready for publication and several questions should be addressed:

- 1 Given that previous studies, including the examples cited by the authors themselves, have primarily focused on cobalt spinels, what advantages does CoOOH offer compared to Co₃O₄?
- 2 The author posits in the text that the OER mechanism of CoOOH is OPM, but relying solely on KIE is insufficient. The author needs to conduct detailed characterizations of the synthesized CoOOH to substantiate the OPM pathway. Please

refer to the methods of the literature Nat Catal 4, 1012–1023 (2021).

3 In Figure 3a, the specific meanings of the curves and points are not clearly labeled by the author. Additionally, the method for calculating the redox charge should be clarified. Please specify the integration formula used and how the peak range and peak baseline are defined.

4 The PEM performance described in this manuscript is not particularly outstanding. On page 17, line 35, it states, "Overall, CoOOH exhibits relatively stable performance comparable to the La and Mn co-doped Co₃O₄ catalyst (200 mA cm⁻² for 100 h in ref.3) and γ-MnO₂ catalyst (200 mA cm⁻² for ~1000 h in ref.50) in the PEM water electrolyzer." However, stability at different current densities cannot be compared equivalently. The authors should operate at comparable or higher current densities to highlight the performance advantages.

5 Why does the point at which ΔE_{edge} begins to change coincide with the oxidation peak in Figure 2e, whereas in Figure 2f, the point of change occurs much earlier than the oxidation peak?

6 The manuscript contains numerous detail errors. For instance, on page 14, line 15, it mentions "Besides the LOM mechanism," yet the LOM mechanism has never been discussed in the text; the discussion has consistently been about OPM. Additionally, the manuscript fails to indicate which statement corresponds to Figure 1d. These errors create significant reading obstacles and confusion for the reviewer. A thorough revision of the entire manuscript is necessary to address these concerns.

Reviewer #3

(Remarks to the Author)

This manuscript reports the development of efficient CoOOH electrocatalysts for water oxidation in proton exchange membrane (PEM) water electrolyzers. The authors investigated the active state of the catalysts using electrochemical methods and operando XAFS techniques. They propose that the Co valence state depends on the electrolyte pH and that the oxygen evolution reaction (OER) in CoOOH proceeds via an oxide path mechanism. However, considering that Nature Communications is a high-impact journal, the reaction mechanism should be investigated from multiple perspectives to ensure a more comprehensive understanding. In its current form, this manuscript does not meet the standards expected for publication in Nature Communications. I recommend submitting it to a more specialized journal. Below are specific points related to the XAFS analysis that should be revised or considered further:

1) While the authors present a b-CoOOH reference in Supplementary Figure 3, a comparison with a more general CoOOH reference should also be included in Figure 1 for both XRD and XAFS. CoOOH can be synthesized using various methods, and previous studies (e.g., those by Prof. Nocera) have reported efficient OER activity of CoOOH structures. What differentiates the CoOOH catalyst prepared in this work from conventional CoOOH materials?

2) If the crystallinity of the CoOOH catalyst in this study is unique, various CoOOH catalysts with different crystallinities should be synthesized. The authors should investigate the relationship between catalytic activity and crystallinity as determined by XRD and EXAFS.

3) In Figure 1d, the FT-EXAFS analysis should be extended to a longer distance region. The discussion of second-nearest-neighbor Co atoms would be useful for evaluating the local cluster size.

4) How was the wavenumber space weighted in the EXAFS analysis? The manuscript should specify whether a k³-weighted or k²-weighted approach was used.

5) In Supplementary Table 2, the coordination numbers for Co-O and Co-Co should be fixed at 6.0. Additionally, the error values for the Debye-Waller factor should be provided.

6) The authors suggest that the Co valence increases during OER, as observed in operando XANES measurements (Figure 2c). If this is the case, a similar shift should also be observed in operando Co L-edge XAFS. Operando soft X-ray XAFS is now widely available and should be included to support this claim.

7) I do not fully agree with the authors' assertion that "only a thin surface layer of CoOOH is reconstructed and oxidized into Co(IV) species under OER conditions." To verify this claim, CoOOH samples with different thicknesses should be synthesized, and the relationship between Co valence and sample thickness should be evaluated. If the CoOOH layer is sufficiently thin, would a larger shift in Co valence be observed?

8) The relationship between Co valence and the half-edge energy (the energy at half the edge-jump intensity) should be examined. Using Co(OH)₂ as a Co²⁺ reference and CoOOH as a Co³⁺ reference could provide a more quantitative estimate of the Co valence state.

9) The photograph of the flow cell in Supplementary Figure 5 is unclear. A schematic diagram should be provided instead. Additionally, the significance of this measurement setup should be clarified, as it appears to be a standard operando XAFS cell.

Version 1:

Reviewer comments:

Reviewer #1

(Remarks to the Author)

All concerns related to my questions have been addressed appropriately. Therefore, I recommend acceptance of this manuscript for publication in Nature Communications.

Reviewer #2

(Remarks to the Author)

In this revised version, the authors have addressed most of the questions effectively. However, some issues remain unanswered due to a lack of experimental evidence. Additionally, certain details could potentially mislead readers or create ambiguity. In my opinion, the scientific rigor and robustness of this manuscript still require further improvement. Therefore, I cannot recommend this version for publication in its current state.

1. In response to Comment 3, the authors attributed the absence of FT-IR experiments to instrument malfunction—an explanation that appears unsubstantiated. Furthermore, the DEMS results failed to demonstrate clear ^{18}O detection signals due to reportedly low intensity. Does this imply that the material in this paper does not follow the OPM pathway? Given the extensive use of DEMS (J. Am. Chem. Soc. 2023, 145, 7829–7836 (2023); Adv. Mater. 37, 2411709 (2024).) in validating OPM mechanisms in prior studies, the authors cannot dismiss this issue perfunctorily. While the OPM mechanism is relatively novel compared to AEM and LOM, and could significantly enhance the scientific depth of this work, the current lack of compelling evidence weakens its credibility.

2. The authors mentioned that “There is only one discernable redox pair observed in the acidic electrolyte, in contrast to the two pairs of redox peaks that are assigned to Co II/III and Co III/IV redox in alkaline and quasi-neutral electrolytes”. In fact, at pH=7 in Figure 2a, no distinct oxidation peak was observed. At pH=1, a reduction peak appeared around a potential of 1.67 V. Therefore, the CV patterns under neutral and acidic conditions are similar, which seems to contradict the author's claims.

3. How do you define the corresponding formal redox potential (denoted as $V_o(\text{Co II/III})$)? In Supplementary Figure 2c, the max potential of oxidation peak is close to 1.6 V, which seems to contradict the value in Fig. 2b.

4. The Mott-Schottky analysis the author used should be clarified in details (e.g. measurement methods, equation and parameters).

5. This work appears to bear similarity to another paper “Oxidation of interfacial cobalt controls the pH dependence of the oxygen evolution reaction” published in Nature Chemistry, with research object, study methods, and some conclusions seemingly overlapping. It creates confusion regarding the novelty of the findings presented in this manuscript.

Reviewer #3

(Remarks to the Author)

The authors have thoroughly addressed my comments and provided appropriate clarifications supported by additional data and thoughtful discussion. The manuscript has been substantially improved and now presents a clearer and more comprehensive understanding of the OER activity and mechanism of CoOOH under acidic conditions. Therefore, I support the acceptance of this manuscript for publication in Nature Communications.

Version 2:

Reviewer comments:

Reviewer #2

(Remarks to the Author)

The author's reply is very nice. I have no more questions and agree to recommend for acceptance.

Response Letter of

Directly synthesized cobalt oxyhydroxide as an oxygen evolution catalyst in proton exchange membrane water electrolyzers (NCOMMS-25-06391-A)

Dear Reviewers,

We sincerely appreciate your help in providing the professional comments/suggestions that aim to improve our manuscript. We tried our best to perform new experiments and supplement new data to improve the rigor of the scientific discussion. We hope that your concerns have been adequately addressed. The point-by-point responses to each comment/suggestion are outlined in the following. The revisions made in the main text and in the Supplementary Information are highlighted in yellow.

Contents

Reviewer #1:	2
Comment 1.....	2
Comment 2.....	4
Comment 3.....	5
Comment 4.....	6
Comment 5.....	7
Comment 6.....	8
Reviewer #2:	9
Comment 1.....	9
Comment 2.....	10
Comment 4.....	14
Comment 5.....	15
Comment 6.....	17
Reviewer #3:	17
Comment 1.....	18
Comment 2.....	20
Comment 3.....	21
Comment 4.....	23
Comment 5.....	23
Comment 6.....	24
Comment 7.....	25

Comment 8.....	26
Comment 9.....	29

Reviewer #1:

The authors have conducted a thorough investigation into the pH-dependent OER performance of CoOOH, demonstrating its potential for practical applications in PEM water electrolysis. To explore the underlying factors affecting the OER activity of CoOOH in electrolytes of different pH values, the authors employed electrochemical analysis combined with operando XAS characterizations, providing in-depth insights through data analysis. Furthermore, DFT calculations have verified the oxide path mechanism of CoOOH, which aligns well with the experimental observations. Overall, the manuscript is of high quality in terms of scientific significance, data analysis, and writing style. Therefore, I recommend its acceptance for publication after addressing the following questions.

Response: We sincerely thank you for your great accomplishment on our manuscript. We will try our best to answer your questions in the following.

Comment 1.

Co₃O₄ typically transforms into CoOOH under OER conditions. Based on this, does Co₃O₄ exhibit a similar pH-dependent OER performance as CoOOH?

Response: Thank you for your agreement that the transformations into CoOOH are necessary for catalyzing the OER in Co-based catalysts. There is no doubt that Co₃O₄ will also exhibit a similar pH dependent OER performance as CoOOH. To support this, we have performed the electrochemical measurement on a Co₃O₄ control sample in alkaline and acidic electrolytes (i.e., pH = 13 and 1, respectively), as shown in New **Supplementary Figure 4** reproduced in the following.

Obviously, the Co redox peaks in the Co₃O₄ control sample are also anodically shifted when decreasing the electrolyte pH, similar to those in the CoOOH (in **Figure 2a**). More importantly, the pH-induced shifts in ^{CoII/III} and ^{CoIII/IV} redox peaks are different, we will discuss this in detail in our **Response to Next Comment 2**.

Furthermore, the other very important consideration is the redox charges (Q_{redox}) extracted from the CV area ($Q_{\text{redox}} = \text{Area}/v$, where v is the scan rate in V s^{-1} , inset in **Supplementary Figure 4a**). The redox charges are obviously reduced in a lower pH environment. This is consistent with our previous findings that the Co^{II} species are difficult to oxidize into Co^{III} species in acidic electrolytes (Nat Commun 15, 3067 (2024)). As a result, the

surface reconstruction of Co_3O_4 into CoOOH is pH-dependent, since the different redox charges clearly indicate the abundance of oxidized Co species changes with the electrolyte pH. This finding is also consistent with the results in our recent publication (*Nat. Chem.* (2025).), cited as new **ref. 24** in this revised version), which clearly point out that the Co oxidation changes prior to the OER onset (i.e., related the redox-active Co species) are smaller in acidic than in alkaline environments. As expected and similar to the trend seen in CoOOH , the OER activity of the Co_3O_4 control sample is also lower in acidic environments compared to alkaline environments, as indicated by the Tafel analysis (**Supplementary Figure 4b**).

Comparison of (a) CV curves and (b) Tafel plots in the electrolytes of pH = 13 and 1. The inset

Supplementary Figure 4. The pH-dependent Co redox process and OER activity. in (a) shows the redox charges (Q_{redox}) extracted by integrating the area of the corresponding CV curves ($Q_{\text{redox}} = \text{Area}/v$, where v is the scan rate in V s^{-1}).

Based on these new results, we have added the related discussions on page 5 line 15 of the main text:

“In addition, a similar super-Nernstian shift in the redox peaks is also evidenced in a Co_3O_4 control sample (**Supplementary Figure 3-4**), as its surface will evolve into CoOOH under OER conditions.”

and on page 8 line 13 of the main text:

“In comparison, the pH-induced change in redox charges is more pronounced in the Co_3O_4 control sample (**Supplementary Figure 4**), which could be explained by the unfavorable Co^{II} to Co^{III} oxidation that occurs in acidic environments, but not in alkaline environments^{6, 24}.”

To make the results in Supplementary Information more consistent and readable, the structural characterizations of this Co_3O_4 control sample are now moved upward as the **New Supplementary Figure 3**.

Comment 2.

In Figure 2a, why is only one discernible redox pair observed in acidic electrolyte, whereas two pairs of redox peaks are observed in alkaline and quasi-neutral electrolytes? Additionally, does the redox pair observed in the acidic electrolyte correspond to the $\text{Co}^{\text{III/III}}$ or $\text{Co}^{\text{III/IV}}$?

Response: Thank you for the comment. First, we would like to point out that the one redox pair observed in the acidic environments represents the combined features of the overlapping $\text{Co}^{\text{II/III}}$ and $\text{Co}^{\text{III/IV}}$ redox processes.

Then, we want to explain the origin for this overlap. Note that, both $\text{Co}^{\text{III/III}}$ or $\text{Co}^{\text{III/IV}}$ redox peaks are anodically shift on the RHE scale, indicating the occurrence of a super-Nernstian shift, i.e., the numbers of electrons and protons transferred during the Co redox process are not equal. In fact, this super-Nernstian shift is commonly found in Co-based catalysts, and usually it is more significant in the $\text{Co}^{\text{III/III}}$ redox peaks (referred to **Ref. 23-24**, Nat. Catal. 5, 414-429 (2022), and Nat. Chem. (2025)). In our new **Supplementary Figure 4**, by measuring the Co_3O_4 control sample, we also show that the super-Nernstian shift is more significant in the $\text{Co}^{\text{III/III}}$ redox peaks compared to the $\text{Co}^{\text{III/IV}}$ redox peaks; however, two redox pairs are still identifiable in acidic environments. For CoOOH herein, the significant super-Nernstian shift in the $\text{Co}^{\text{III/III}}$ redox peaks leads to an overlap with the $\text{Co}^{\text{III/IV}}$ redox peaks, resulting in only one discernable broad redox pair.

Following your comments, we believe further clarification will help to reduce confusion in the readers. Therefore, we have reconstructed the related discussion on **page 5 line 12** of the main text as follows:

“... The CVs recorded at different pHs show a clear super-Nernstian shift in the Co redox peaks, as the numbers of electrons and protons transferred during the Co redox processes are not equal, which is commonly found in Co-based catalysts^{23, 24}. In addition, a similar super-Nernstian shift in the redox peaks is also evidenced in a Co_3O_4 control sample (**Supplementary Figure 3-4**), as its surface will evolve into CoOOH under OER conditions. For CoOOH , the super-Nernstian shift is more significant in the $\text{Co}^{\text{III/III}}$ redox peaks, causing them to overlap with the $\text{Co}^{\text{III/IV}}$ redox peaks in an acidic environment.”

Comment 3.

Under OER conditions, surface reconstruction from CoOOH to the active phase CoO₂ typically occurs. To improve the accuracy of the theoretical calculations, it would be beneficial to use CoO₂ as the model for DFT calculations.

Response: Thank you for the suggestion. In addition to the presented DFT calculations, we would also have performed the DFT calculations as suggested if we had been able to find structural information for CoO₂ at the beginning. In addition, we want to draw your attention to the fact that the existence of CoO₂ or even Co^{IV} is still under discussion. On the one hand, the oxidative transition of 2+→3+→4+ is widely accepted for Co-based catalysts in the literature. On the other hand, it is still challenging to really detect Co⁴⁺ under OER conditions. For example, by tracking the operando XAS at the Co-L₃ edge, Prof. William C. Chueh and coworkers suggested that Co is oxidized into Co³⁺ under OER conditions, without obvious further oxidation into Co⁴⁺ (Nature 593, 67–73 (2021)). In addition, Prof. Beatriz Roldan Cuenya and coworkers suggested that oxidized Co species above Co³⁺ can be denoted as Co^{3+δ} (referred to Ref. 30, Nat. Energy 7, 765-773 (2022)).

Therefore, on page 7 line 20, we clarify that: “Both the bond length and coordination number from fitting remain similar at different currents (Supplementary Figure 9), suggesting that only a thin surface layer of CoOOH is reconstructed and oxidized into Co^{IV} species¹⁵ (or Co^{3+δ} referred to ref. 30) under OER conditions.”

On page 7 line 34, we clarify that: “Therefore, though the low-spin Co^{III} species at the surface of CoOOH can be further oxidized into Co^{IV} species (or Co^{3+δ} referred to ref. 30) under OER conditions, they reversibly return back to the initial state when CoOOH is taken out of the electrolyte and dried.”

As far as we know, the formation of CoO₂ is mostly supported by in-situ Raman characterization, and the structural information provided is insufficient. Therefore, we hope to have your understanding that we are unable to perform the suggested DFT calculations at this stage. The model for the DFT calculations done in this manuscript is built according to the paper published by Prof. Jens K. Nørskov, Prof. Alexis T. Bell and coworkers, who are well-known experts in the field (referred to Ref. 44, J. Am. Chem. Soc. 135, 13521-13530 (2013)). We have adapted a similar model as used in this paper to improve the accuracy of the theoretical calculations. Furthermore, we have also clearly pointed out the gaps between the DFT calculations and experimental results with systematic discussion in the whole paragraph, beginning on page 14 line 7.

Generally, narrowing the gap between the DFT calculations and the experimental results are one of the biggest challenges in this field. We are sorry that it is unlikely to be resolved in this manuscript. With all the efforts and clarifications in this manuscript, we tried our best to avoid misleading the readers.

Comment 4.

Does CoOOH fully follow the OPM pathway, or is it partially following this mechanism? An isotope labeling DEMS experiment could provide additional insights into the OER pathway.

Response: Thank you for this professional comment that reminds us to double check the discussion about the reaction mechanism. We appreciate your nice suggestion and agree with you that an isotope labelling DEMS experiment would be helpful to provide additional insights into the OER pathways. So far, we have already performed the experiment of deuterium substitution in the electrolytes to reveal whether a proton is involved in the rate-determining step (detailed in **Figure 3d-e**).

However, to determine the reaction pathway for a complex multiple-step reaction like the OER really requires a precise probe of the electrocatalytic surface at the molecular level and with time-resolution, which is still challenging to date. As far as we know, the recommended method of isotope labelling/substitution does not enable a direct probe of the electrocatalytic surface. It is difficult to differentiate the lattice oxygen mechanism (LOM) from the OPM pathway, especially when metal dissolution happens in acidic environments for Co-based catalysts. Then, the interpretation of the experimental results is based on the preset models (e.g., LOM vs. OPM pathways) with its limitations. Similarly, when we discuss the results of deuterium substitution in the section of “**Proton transfer during OER**” (beginning on **page 10 line 30** of the main text), we point out that proton is not involved in the rate-determining step. However, we also need to make clear clarifications and be conscious of the limitations, to avoid over-interpretation.

Following your suggestion, we have also systematically checked the ^{18}O -labelling DEMS experiments done in the literature. We find that labelling of the sample is most challenging, and it affects the measured results and their interpretation. More importantly, the proportion of the labelled ^{18}O detected in the product is generally low (potentially as low as below 1%) compared to the proportion of ^{16}O , meaning it is difficult to provide valuable information from which an effective interpretation can be made. Herein, we also want to draw your attention to the stability issue of CoOOH and Co-based catalysts (J. Am. Chem. Soc. 2025, 147, 4, 3517–3528), which will challenge the ^{18}O -labelling of the studied samples and

the interpretation of measured results, due to the complicated simultaneous processes of Co dissolution and OER at the surface.

After careful evaluation and reconsidering the time and scope of this publication, we hope to have your understanding that the ^{18}O -labelling DEMS experiments could be helpful but cannot provide an unquestionable understanding of the reaction mechanism, which requires a direct probe of the electrocatalytic surface at the molecular scale and with time resolution. Inspired by your comment, we do realize further clarifications on the related discussion is needed. We have reformed the discussion on page 14 line 18 of the main text:

“... Besides the OPM mechanism^{5, 7} proposed here, other mechanisms have also been proposed for the CoOOH and Co-based catalysts^{14, 23, 48}. In addition, the AEM pathway can also co-exist with the OPM pathway⁵ or other pathways. Indirect probe methods at this stage make it difficult to capture the dynamic and complex interfacial electrochemical processes at the molecular scale required to determine the reaction pathways. Therefore, from experimental point of view, further efforts are required to precisely elucidate the complex transformation of interfacial intermediates under OER conditions.”

In addition, we have also reformed the discussion in the **Abstract** and **Conclusion and Outlook** sections.

On page 1 line 26:

“In addition, we find that there is no primary proton/deuterium (H/D) kinetic isotope effect (KIE) in acidic environments, and the apparent activation energy (E_{app}) at the onset of OER decreases with pH, aligning well with an oxide path mechanism (OPM) as the dominated path for the OER in CoOOH.”

On page 19 line 3:

“The oxide path mechanism (OPM) was verified by DFT calculations to align well with the experimentally observed pH-dependent trend of apparent activation energy (E_{app}) and absence of a primary H/D KIE. However, other reaction pathways cannot be excluded at this stage. The direct probe method, which allows operando tracking of the electrochemical surface at the molecular scale and with time resolution, will be beneficial to further clarify the OER pathways in the future.”

Comment 5.

After the acidic OER, the Co oxidation state in CoOOH increases (as shown in Figure 5a). The impact of this change on the surface structure of catalyst should be considered.

Response: Thank you for the comment. The change in Co oxidation state results from continuous Co oxidation in the sublayer. We have continuously found this phenomenon in many different catalysts (Nat. Chem. (2025), Adv. Energy Mater. 2024, 14, 2303529, Energy Environ. Mater., 7: e12737, etc.). Recently, Prof. Peter Strasser and coworkers also suggested the oxidation of the catalyst goes beyond just the surface (Nat Catal 7, 1213–1222 (2024)). However, we note that these oxidized species can be reversed back to the Co^{III} state with the ex-situ soft XAS characterizations (detailed in **Figure 2g-h**). To address your concern and further clarify the oxidation of CoOOH under OER conditions, we have added the following discussion on **page 14 line 32** of the main text:

“At the end of the third CV cycle, a change of 60 meV of the ΔE_{edge} arises from the formation of thicker activated layers at the surface, which is commonly observed since Co oxidation in the sublayer is coupled with the formation of a surface layer that directly participates in the OER²⁴. However, these oxidized layers may reverse back to Co^{III} state when the catalyst is taken out from the electrolyte and dried, as evidenced by the ex-situ soft XAS characterizations (**Figure 2g-h**).”

Comment 6.

While the oxygen evolution activity of CoOOH shows significant pH dependence, does the stability of CoOOH also exhibit a similar dependence on pH?

Response: Thank you for the comment. CoOOH is a well-known OER catalyst in alkaline and neutral environments and shows good stability. Therefore, in this manuscript, we try to showcase that CoOOH is also quasi-stable in acidic environments, though dissolution of Co is inevitable under these conditions, as detailed in **Supplementary Table 5**. Further structural engineering is required to improve its stability, as we addressed in the “**Conclusions and Outlook**” section (**page 18** of the main text).

The pH-dependence of CoOOH is more evident in the Pourbaix diagram, as the following **Figure R1** adapted from the papers by published by Prof. Jens K. Nørskov, Prof. Alexis T. Bell and coworkers (referred to **Ref. 44**, J. Am. Chem. Soc. 135, 13521-13530 (2013)). Following your comment, we have clarified this point by adding a sentence on **page 15 line 19** of the main text:

“According to the Pourbaix diagram, the Co is less stable in an acidic environment rather than in neutral or alkaline environments⁴⁴. The dissolution of Co in the acidic electrolyte after the CP measurement was quantified by inductively coupled plasma optical emission spectroscopy (ICP-OES, **Supplementary Table 5**).”

Figure R1. Pourbaix diagram of Co, which is adapted from J. Am. Chem. Soc. 135, 13521-13530 (2013) (original Figure 1a in the reference paper) and will not be included in the final publication.

Reviewer #2

Comments on “Directly synthesized cobalt oxyhydroxide as an oxygen evolution catalyst in proton exchange membrane water electrolyzers”

In this manuscript, the authors demonstrate that the well-known CoOOH catalyst shows promising activity and stability toward the OER in acidic environments. Although the author conducted extensive characterizations and electrochemical analyses to investigate the mechanism of CoOOH, the driving force and innovation of this research appear to be somewhat lacking. This article is still far from being ready for publication and several questions should be addressed:

Response: Thank you for taking the time to review our manuscript. We appreciate your feedback and have made the following revisions based on your comments and suggestions. With the discussion below, we hope to further share our understanding of the driving force and innovation behind this manuscript.

Comment 1.

Given that previous studies, including the examples cited by the authors themselves, have primarily focused on cobalt spinels, what advantages does CoOOH offer compared to Co₃O₄?

Response: From the above comment “*Given that previous studies....have primarily focused on cobalt spinels*”, we thank you for your recognition of the important research on CoOOH for

PEM water electrolyzer done in this manuscript, which is the novelty compared to those reported in the literature about cobalt spinels. In **Figure 6c**, we proved that CoOOH shows a similar performance to another two Co oxide samples (including the Co₃O₄ control sample) in PEM water electrolysis. The results are reasonable, considering Co₃O₄ needs to reconstruct into a CoOOH-like structure under OER conditions. Therefore, only from the perspective of device performance, the CoOOH and Co₃O₄ are competitive.

However, little is known about the behavior of CoOOH in acidic environments. Therefore, the most important driving force for us to start this research is to prove that the CoOOH structure is quasi-stable for the OER in acidic environments, which has not yet been done in the literature. Surface reconstruction is an inevitable topic for OER catalysts. I hope you may agree *that the research done with CoOOH in this manuscript can fill the gap between the initial Co₃O₄ structure and a final Co⁴⁺ (or Co^{3+δ}) OER-active state*, which is important for the future research in the field, from both experimental and theoretical perspectives.

More importantly, differently from the spinel structure, the CoOOH has a layered structure that can enable different structural engineering strategies which boost the electrocatalytic performance, broadening the choices for the catalyst design and inspiring further research on optimizing its OER performance in acidic environments. On this basis, and with your help, we hope to introduce to the research community the new role of the already well-known CoOOH as a promising OER catalyst in PEM water electrolysis, or other PEM related electrochemical conversions.

Comment 2.

The author posits in the text that the OER mechanism of CoOOH is OPM, but relying solely on KIE is insufficient. The author needs to conduct detailed characterizations of the synthesized CoOOH to substantiate the OPM pathway. Please refer to the methods of the literature *Nat Catal* 4, 1012–1023 (2021).

Response: Thank you for this insightful suggestion. In the recommended paper (Nat Catal 4, 1012–1023 (2021)), Lin et al. have applied both the isotope labelling DEMS and in-situ FT-IR characterization to prove the occurrence of an OPM pathway in the Ru/MnO₂ catalyst. We notice **Reviewer #1** also suggested a similar method of using isotope labelling DEMS to provide new insights into the reaction mechanism. Therefore, we believe that the discussion about the OER mechanism is of interest to research community, since it is still very challenging to determine the reaction pathway for OER catalysts.

In fact, we did plan to perform in-situ FT-IR characterizations on the CoOOH catalyst at the beginning since, as far as we know, it is the most surface-sensitive technique to unveil the adsorbate species existing before the RDS (i.e., in the resting state). That is also the reason that we cited the recommended paper (Nat Catal 4, 1012–1023 (2021)) as **Ref. 45** in the previous version of our manuscript, since the research done there is a very good demonstration of the OPM discussion. Unfortunately, we ran into an unfavorable situation, due to a non-ideal cell design leading to a leakage of the electrolyte inside the vacuum chamber, ruining our FT-IR spectrometer.

In addition, we have also systematically checked the ^{18}O -labelling DEMS experiments done in the literature. We find that labelling of the sample is the most challenging aspect, and the proportion of the labelled ^{18}O detected in the product is generally very low compared to that of ^{16}O (potentially as low as below 1%), meaning it is difficult to make a strong interpretation of the ^{18}O -labelling results. For example, in a well-recognized paper in the field (Nat. Chem. 9, 457–465 (2017), cited as the New **Ref. 48**), Prof. Yang Shao-Horn and coworkers also detected a $^{36}\text{O}_2$ signal in the Co-based catalysts that would be generated by a proposed lattice oxygen mechanism (LOM), instead of the OPM proposed here and in the recommended paper (Nat Catal 4, 1012–1023 (2021)). Importantly, the CoOOH studied here is only meta-stable, which will challenge the ^{18}O -labelling of sample and the interpretation of the measured results, due to the complex simultaneous processes of Co dissolution (accompanied by the loss of lattice oxygen) and the OER at the surface.

After careful evaluation and reconsidering the time and scope of this publication, we hope to have your understanding that the recommended characterizations could be helpful but cannot be done timely and would potentially not help to achieve the goal of clarifying the reaction pathways, which are still under discussion in the literature. This topic remains the most challenging in the field, due to the lack of direct probe studies on the electrocatalytic surface at the molecular scale and with time resolution. Other than that, we tried our best to perform operando hXAS characterizations (**Figure 3b-c and 5**), which is the key strength of our group, to understand the Co oxidation process and its structural stability in CoOOH. We also performed analysis on the H/D kinetic isotope effect and pH-dependent apparent activation energy (**Figure 3d-e and 4a-b**), to prove that the proton and electron transfers should be decoupled in the RDS.

To avoid misleading the readers, we decided to reframe the discussion on page 14 line 18 of the main text:

““Besides the OPM mechanisms^{5, 7} proposed here, other mechanisms have also been proposed for the CoOOH and Co-based catalysts^{14, 23, 48}. In addition, the AEM pathway can

also co-exist with the OPM pathway⁵ or other pathways. Indirect probe methods at this stage make it difficult to capture the dynamic and complex interfacial electrochemical processes at the molecular scale required to determine the reaction pathways. Therefore, from experimental point of view, further efforts are required to precisely elucidate the complex transformation of interfacial intermediates under OER conditions.”

In addition, we have also reformed the discussion in the **Abstract** and **Conclusion and Outlook** sections.

On page 1 line 26:

“In addition, we find that there is no primary proton/deuterium (H/D) kinetic isotope effect (KIE) in acidic environments, and the apparent activation energy (E_{app}) at the onset of OER decreases with pH, aligning well with an oxide path mechanism (OPM) as the dominated path for the OER in CoOOH.”

On page 19 line 3:

“The oxide path mechanism (OPM) was verified by DFT calculations to align well with the experimentally observed pH-dependent trend of apparent activation energy (E_{app}) and absence of a primary H/D KIE. However, other reaction pathways cannot be excluded at this stage. The direct probe method, which allows operando tracking of the electrochemical surface at the molecular scale and with time resolution, will be beneficial to further clarify the OER pathways in the future.”

Comment 3.

In Figure 3a, the specific meanings of the curves and points are not clearly labeled by the author. Additionally, the method for calculating the redox charge should be clarified. Please specify the integration formula used and how the peak range and peak baseline are defined.

Response: Thank you for bringing these issues to our attention. In the caption of **Figure 3a**, we have provided details about the scatter and curves on page 9 line 1 of the main text: “(a) Redox charges (the scatters) estimated by the area of CVs collected at different scan rates and in different electrolytes in **Supplementary Figure 2**. The distribution of the redox charges is indicated by the curve above the scatters.”

In addition, we want to point out that, first, the Co redox processes overlap with the OER. Second, the interfacial capacitance of OER catalysts is potential-dependent (referred to ACS Catal. 2019, 9, 10, 9222–9230, and ACS Catal. 2023, 13, 2, 1186–1196). Under these circumstances, it difficult to define a baseline. Therefore, we decided to integrate the area of the CV to calculate the total charge to represent the redox charge (i.e., oxidation charge +

reduction charge) of Co species. As the total charge has other contributions (e.g., OER and double layer capacitance), it is slightly higher than the actual redox charge.

Supplementary Figure 2e. (e) Scheme showing the calculation of redox charge (Q_{redox}) based on the CV area ($Q_{\text{redox}} = A/v$, where A is the area of the CV and v is the scan rate in V s^{-1}). The CV area is obtained by integration. Due to the overlap of the OER with the Co redox processes, we note that there are contributions from the charge from the OER, and also from the charge stored in the electric double layers. Therefore, the total charge calculated by the area of the CV is slightly higher than the charge passed during the interfacial redox (oxidation + reduction) processes.”

To avoid confusing the readers, we decided to first add a scheme as the new **Supplementary Figure 2e** to show how to calculate the redox charge. Second, we reformed the related discussion on page 8 line 4 of the main text:

“The redox charge passed during the interfacial oxidation and reduction of Co species can be estimated by calculating the total charge from the area of CVs (see **Supplementary Figure 2** for more details). Due to the overlap of the OER with the Co redox processes and also the contributions from double layer capacitance, the as-calculated redox charge is slightly over-estimated. To reduce these influences, the redox charges are extracted from CVs collected at

different scan rates, and their values are summarized in **Figure 3a**. Although the formal redox potential shifts significantly as a function of the electrolyte pH, the estimated average redox charges extracted in different electrolytes remain quite constant at around 140 to 150 C g^{-1} , suggesting that the coverage of redox-active species in CoOOH is only slightly affected by the pH environment.”

Comment 4.

The PEM performance described in this manuscript is not particularly outstanding. On page 17, line 35, it states, "Overall, CoOOH exhibits relatively stable performance comparable to the La and Mn co-doped Co₃O₄ catalyst (200 mA cm⁻² for 100 h in ref.3) and γ-MnO₂ catalyst (200 mA cm⁻² for ~1000 h in ref. 50) in the PEM water electrolyzer." However, stability at different current densities cannot be compared equivalently. The authors should operate at comparable or higher current densities to highlight the performance advantages.

Response: Thank you for the comment. We have further performed a stability measurement at 200 mA cm⁻² to see the stability of CoOOH, as shown in new **Supplementary Figure 36** in the following. The applied potential increases slowly in the first ~100 h, before increasing significantly after that. The change in darkness of the CCM can be also identified after the CP test, due to CoOOH dissolution. Therefore, the CoOOH indeed suffers from Co dissolution when further increasing the current density higher than above 100 mA cm⁻².

To avoid misleading to the readers, we have reconstructed the related discussion on **page 18 line 15** of the main text:

"... In contrast, when the applied current density was set to 200 mA cm⁻² and 500 mA cm⁻² for the CP measurement, the change in cell potential is significant due to Co dissolution (**Supplementary Figure 36-38**).

Overall, CoOOH exhibits relatively stable performance at the lower current density of 100 mA cm⁻², compared to a La and Mn co-doped Co₃O₄ catalyst (200 mA cm⁻² for 100 h in ref.³) and a γ-MnO₂ catalyst (200 mA cm⁻² for ~1000 h in ref.⁵⁴) in the PEM water electrolyzer."

Supplementary Figure 36. CP measurement at 200 mA cm⁻². (a) The CP curve recorded at a constant current density of 200 mA cm⁻² for 130 h. The inset pictures show the CCM at the

CoOOH side before and after the CP test; a decrease in the darkness of the CCM indicates the dissolution of CoOOH under the condition of 200 mA cm⁻².

Comment 5.

Why does the point at which ΔE_{edge} begins to change coincide with the oxidation peak in Figure 2e, whereas in Figure 2f, the point of change occurs much earlier than the oxidation peak?

Response: Thank you for this professional comment. In the past few years, we have consistently revealed that the onset of ΔE_{edge} from the operando XAS characterizations coincides with the flat band potential (V_{fb}) of Co-based catalysts; in addition, the flat band potentials are also usually located close to the $\text{Co}^{\text{II/III}}$ redox potentials (please check these papers for details if you are interested in this topic: Nat. Chem. (2025); Adv. Energy Mater. 2024, 14, 2303529; Adv. Funct. Mater. 2024, 34, 2405384).

To address your comment, we have performed an EIS characterization and Mott-Schottky analysis to extract the V_{fb} for CoOOH in alkaline and acidic electrolytes, as shown in the new **Supplementary Figure 11-12**. Consistently, we found that the onset of the ΔE_{edge} in CoOOH is also located at around the V_{fb} extracted from Mott-Schottky analysis. We note that the slight mismatch between these values could result from limitations in the different techniques used to extract these parameters, as well as the different interfacial properties present in different electrocatalytic systems.

To clarify the connection between the onset of the ΔE_{edge} and V_{fb} , we have added the following statement on page 7 line 15 of the main text:

“In addition, the onsets of ΔE_{edge} (i.e., ~ 1.20 and ~ 1.40 V vs. RHE in pH = 1 and 13, respectively) is also located at around the flat band potential of CoOOH (1.15 ± 0.05 and 1.51 ± 0.07 V vs. RHE in pH = 1 and 13, respectively, **Supplementary Figure 11-12**),, which is consistently observed for Co-based catalysts^{24, 26, 29}.”

To explain the gap between different parameters, the related discussion has been added after the **Supplementary Figure 12**:

Discussion: For Co-based catalysts, the V_{fb} determined by Mott-Schottky analysis is frequency dependent and is usually located at around the Co redox potentials. More importantly, the onset of the ΔE_{edge} is consistently located close to the V_{fb} (ref.^{3, 4, 5}). Here, we also show that the onset of the ΔE_{edge} is close to the V_{fb} for CoOOH in both alkaline and acidic electrolytes. The consistent observations could be due to the fact that these three parameters, namely, the onset of ΔE_{edge} , V_{fb} , and $V^{\circ}(\text{Co}^{\text{II/III}})$ are closely correlated with the interfacial Co oxidation in Co-based catalysts. However, slight deviations in these parameters and their pH-

dependence could result from limitations of the different techniques used to extract these parameters, as well as the varied interfacial properties in different electrocatalytic systems.”

Supplementary Figure 11. EIS plots at different potentials. EIS characterization to extract the flat band potentials for CoOOH (a) in 0.1 M KOH with pH = 13 and (b) in 0.05 M H_2SO_4 with pH = 1. The insets in (a-b) show the magnification in the lower $\text{Re}(Z)/-\text{Im}(Z)$ regions.

Supplementary Figure 12. Flat band potentials for CoOOH in pH=1 and 13 extracted from Mott-Schottky analysis. Mott-Schottky plots used to extract flat band potential (V_{fb}) for CoOOH (a) in 0.1 M KOH with pH = 13 and (b) in 0.05 M H₂SO₄ with pH = 1, at different frequencies from the EIS characterizations. (c) The extracted V_{fb} are plotted as a function of $\log(f(\text{Hz}))$ in alkaline and acidic electrolytes. (d) Comparison on the V_{fb} and $V^{\circ}(\text{Co}^{II/III})$, in alkaline and acidic electrolytes, respectively. The horizontal dashed lines in (c-d) are the onset potentials for ΔE_{edge} determined from operando hXAS characterizations. The coloured lines beside the box in (d) show the distribution of the data points.

Comment 6.

The manuscript contains numerous detail errors. For instance, on page 14, line 15, it mentions "Besides the LOM mechanism," yet the LOM mechanism has never been discussed in the text; the discussion has consistently been about OPM. Additionally, the manuscript fails to indicate which statement corresponds to Figure 1d. These errors create significant reading obstacles and confusion for the reviewer. A thorough revision of the entire manuscript is necessary to address these concerns.

Response: Thank you for bringing these issues to our attention. We sincerely apologize for these irrational mistakes in the manuscript. We have carefully checked through the manuscript and made the following revisions:

On **page 4 line 10** of the main text:

"The corresponding Fourier transformed extended X-ray absorption fine structure (EXAFS) spectrum of CoOOH shows two distinct peaks, corresponding to the Co-O and Co-Co scattering paths, respectively (**Figure 1d**)."

On **page 14 line 18** of the main text:

"Besides the OPM mechanisms^{5, 7} proposed here, we note that other mechanisms has also been proposed for the CoOOH and Co-based catalysts..."

Other minor revisions throughout the manuscripts have also been highlighted in **Yellow**.

Reviewer #3

This manuscript reports the development of efficient CoOOH electrocatalysts for water oxidation in proton exchange membrane (PEM) water electrolyzers. The authors investigated the active state of the catalysts using electrochemical methods and operando XAFS techniques. They propose that the Co valence state depends on the electrolyte pH and that the oxygen evolution reaction (OER) in CoOOH proceeds via an oxide path mechanism.

However, considering that Nature Communications is a high-impact journal, the reaction mechanism should be investigated from multiple perspectives to ensure a more comprehensive understanding. In its current form, this manuscript does not meet the standards expected for publication in Nature Communications. I recommend submitting it to a more specialized journal.

Response: We sincerely thank you for your time and consideration of our manuscript. We totally agree that research from different perspectives will help to comprehensively reveal the reaction mechanism. As a research group that has been focused on the OER for more than 10 years, we are well aware of this importance. However, due to the complexity of surface reconstruction under OER conditions and the lack of sufficient direct probe techniques, discussion about OER mechanism still does not reach a general agreement in the literature and in the community. The situation is even more complicated for Earth abundant OER catalysts in acidic environments. Different reaction mechanisms have been proposed to explain experimental observations that cannot be predicted by a conventional adsorbate evolution mechanism (AEM). However, it is still difficult to differentiate different pathways at the molecule level, preventing to reach a strong conclusion on mechanistic discussion. In this revised manuscript, we tried our best to revise the discussion about OER mechanism to avoid misleading the readers.

Aside from the mechanistic study, for the first time, we have applied directly synthesized CoOOH in a PEM water electrolyzer, which is supported by sufficient fundamental studies on the Co redox processes, interfacial Co oxidation, proton transfer, apparent kinetic effect, DFT calculations, etc. Then, *Nature Communications* is a top open-access and interdisciplinary journal, aiming to narrow the gap between academic knowledge and real-life applications. Therefore, we believe that the technological relevance and potential demonstrated in this manuscript is of interest to its editorial office and to its broader readership. We will address your following comments with new experiments and analyses, to improve the rigor of the related discussions made in this manuscript. We appreciate your further input to review our revisions.

Below are specific points related to the XAFS analysis that should be revised or considered further:

Comment 1.

While the authors present a b-CoOOH reference in Supplementary Figure 3, a comparison with a more general CoOOH reference should also be included in Figure 1 for both XRD and XAFS. CoOOH can be synthesized using various methods, and previous studies (e.g., those

by Prof. Nocera) have reported efficient OER activity of CoOOH structures. What differentiates the CoOOH catalyst prepared in this work from conventional CoOOH materials?

Response: Thank you for your suggestions. We cannot find a commercial CoOOH material online, therefore we have to use commercial Co(OH)₂ from Sigma Aldrich as a starting materials to synthesize b-CoOOH. As clearly written in the **Methods** (page 19 line 34 of the main text), except for the starting materials, other conditions for the synthesis of b-CoOOH are the same as for the CoOOH sample. Therefore, we use the b-CoOOH sample as a general CoOOH reference in this manuscript. In addition, the b-CoOOH is mainly used to prove the consistency of special redox features of the CoOOH structure in acidic environments, which have not been well discussed so far. To avoid confusing the readers, we would like to keep the structural characterizations of b-CoOOH in the Supplementary Information. We have compared the XRD patterns of b-CoOOH with that of CoOOH side by side in the new **Supplementary Figure 5a**, as also shown below.

Supplementary Figure 5a. Comparison of the XRD patterns between b-CoOOH and CoOOH. The full width at half maximum (FWHM) of the XRD pattern for b-CoOOH control sample is smaller than that of the CoOOH sample (Figure 1b), indicating b-CoOOH has better crystallinity.

Thank you for the nice recommendation of the previous studies from Prof. Daniel Nocera (e.g., paper cited as Ref. 21, *J. Am. Chem. Soc.* 2010, 132, 46, 16501–16509). We notice that most of the related research done in his group is about Co phosphate, i.e., CoPi, which is one of the most well-known OER catalysts in neutral environments. Recently, Prof. Holger Dau and coworkers suggested the CoPi could be bulk-active (referred to Adv. Energy

Mater. 2024, 2403818), due to the intercalation of K^+ and phosphate ions between the layers. However, both b-CoOOH and CoOOH studied in this manuscript have the typical beta-phase oxyhydroxide structure, without intercalated ions between layers. To clearly show their differences, we have put schemes of their crystal structures side-by-side in **Figure R2** below. Even though both CoPi in the literature and CoOOH herein are composed of CoO_6 octahedral layers, we would like to point out the CoOOH catalyst studied in this manuscript is actually a **more conventional and representative** catalyst with good crystallinity, compared to well-known amorphous CoPi catalyst in neutral environments. To avoid misleading to the readers, we have clarified this point on page 3 line 27 of the main text:

“The analysis of the XRD pattern reveals that the as-prepared CoOOH is a typical 3R polytype structure¹⁹ with lattice parameters of $a = b = 2.85 \text{ \AA}$, and $c = 13.18 \text{ \AA}$ (**Supplementary Table 1**). There are no intercalated ions between the CoO_6 octahedral layers, different from the noncrystalline layer Co oxide catalysts^{20,21}.”

We are very curious about the performance of CoPi in acidic electrolyte. We will discuss this in detail in our **response to your Comment 7** in the following.

Figure R2. Comparison of the crystal structures for (a) amorphous Co catalysts for neutral OER and (b) the crystalline beta-phase CoOOH studied in manuscript. Panel (a) is adapted from Adv. Energy Mater. 2024, 2403818 to aid the discussion but will not be included in the final publication.

Comment 2.

If the crystallinity of the CoOOH catalyst in this study is unique, various CoOOH catalysts with different crystallinities should be synthesized. The authors should investigate the relationship between catalytic activity and crystallinity as determined by XRD and EXAFS.

Response: Thank you for your comment. We note that the crystallinity of the catalyst should play an important role in the electrocatalytic activity. For example, in **Ref. 54** (Nat Catal 7, 252–261 (2024)), Kong et al. have tuned the crystallinity of the MnO_2 by changing the annealing temperature to optimize the OER performance in acidic electrolytes.

In fact, as the comparison of the XRD patterns in the **Response to Comment 1** above shows, the crystallinity of b-CoOOH is different to the CoOOH catalyst used in this work. Previously, we have compared the CVs of b-CoOOH and CoOOH, and the latter shows a much higher redox current, due to the larger active surface area inherited from the lower crystallinity. In this revision, we have compared the OER performance of b-CoOOH and CoOOH in the new **Supplementary Figure 6**. As reproduced and shown below, the overpotential of CoOOH at a current density of 10 A g^{-1} is reduced by $\sim 30 \text{ mV}$ compared to the more crystalline b-CoOOH, consistent with the general understanding of the relationship between catalytic activity and crystallinity.

Supplementary Figure 6. Electrochemical performance of b-CoOOH. Comparison of (a) the cyclic voltammograms (CVs) and (b) OER polarization curves of the b-CoOOH control sample and the CoOOH sample in 0.05 M H₂SO₄. The CVs were collected at a scan rate of 100 mV s^{-1} . Both samples show only one redox pair in an acidic environment. To avoid the influence from the redox current, the OER polarization curves were collected using the steady state chronoamperometry (CA) technique. A reduction of $\sim 30 \text{ mV}$ in the overpotential at the current density of 10 A g^{-1} is observed in CoOOH when compared to b-CoOOH, consistent with general understanding on the relationship between catalytic activity and crystallinity.

Comment 3.

In Figure 1d, the FT-EXAFS analysis should be extended to a longer distance region. The discussion of second-nearest-neighbor Co atoms would be useful for evaluating the local cluster size.

Response: Thank you for the recommendation. We would like to perform the new fitting as suggested, however, due to the structure of CoOOH, there are several reasons why fitting further is not currently believed to yield reliable information pertaining to the crystallite size. In particular, the number of significant paths required to be fitted to obtain reliable information on the 2nd Co-Co distance, with their own independent Debye Waller factors and distances,

quickly results in a number of parameters exceeding the Nyquist criterion. Qualitatively, plotting the data to a maximum R of 6 Angstrom (as shown in **new Figure 1d** in the following) shows the appearance of clear peaks which can be associated mostly to the 2nd and 3rd Co-Co distances and then further to double and triple scattering pathways. However, there are rather significant double scattering pathways underneath that without proper consideration will significantly hinder refinement of the coordination numbers beyond any reasonable uncertainty that we are satisfied to consider when drawing conclusions. The clear evidence of these peaks alone suggest that the crystallite domains are fairly large, in agreement with the XRD and TEM images, which together all show that the CoOOH has a large size. The scattering paths plotted in **Figure R3** below are only the ones that have a significance factor greater than 10 as determined with FEFF, there are however, several hundred others that can collectively have significant impact further into R space. Further EXAFS analysis here appears to not clearly add additional value.

In general, even though the EXAFS fitting has not been performed to cover a wider range, we did extend the range of radical distance to 6 Angstrom in the new **Figure 1d**, to prove that the crystallite domains of CoOOH are fairly large.

New Figure 1d. Fitting of the k^3 -weighted Fourier transformed EXAFS spectrum at the Co K edge of CoOOH, with the inset showing the layered atomic structure of CoOOH.

Figure R3. The scattering paths for CoOOH structure that have a significance factor greater than 10 as determined with FEFF.

Comment 4.

How was the wavenumber space weighted in the EXAFS analysis? The manuscript should specify whether a k^3 -weighted or k^2 -weighted approach was used.

Response: Thank you for bringing this issue to our attention. We apologize for the omission of this important information for the EXAFS analysis. All EXAFS analyses are performed with a k^3 -weighted approach. We have added this important information in all captions of the EXAFS figures.

Comment 5.

In Supplementary Table 2, the coordination numbers for Co-O and Co-Co should be fixed at 6.0. Additionally, the error values for the Debye-Waller factor should be provided.

Response: Thank you for the recommendation. We have reperfomed the EXAFS analysis without guessing the coordination numbers for the Co-O and Co-Co shells, as shown in the new **Supplementary Table 2** in the following.

Supplementary Table 2. Fitting of the ex-situ EXAFS spectrum

 Fitting parameters of the k^3 -weighted EXAFS spectrum of CoOOH in **Figure 1d**.

	CN	R (Å)	$\sigma^2(\text{Å}^2)$	E_0 (eV)	R-factor
Co-O	6	1.903 ± 0.003	0.003 ± 0.0002	1.173	0.004
Co-Co	6	2.853 ± 0.003	0.004 ± 0.0001		

Note: The fitting was performed with a k range of 2 to 14 Å^{-1} . The coordination number (CN) for each shell was set to be 6. The amplitude reduction factor was set to be 0.80 for the fitting. E_0 refers to the energy shift, R refers to Co-O bond distance, σ^2 is Debye Waller factor and the R-factor is an indicator of fit quality.

Comment 6.

The authors suggest that the Co valence increases during OER, as observed in operando XANES measurements (Figure 2c). If this is the case, a similar shift should also be observed in operando Co L-edge XAFS. Operando soft X-ray XAFS is now widely available and should be included to support this claim.

Response: Thank you for the suggestion about using soft X-ray XAFS to get further evidence. We fully agree with you that the new results would be supportive if it can be done in a convenient way. However, we also want to share our understanding about the limitations of operando soft X-ray characterizations at the current stage. For example, the Co $L_{2,3}$ edge can be collected in the total electron yield (TEY) mode or fluorescence yield (FY) mode. The TEY mode is much more surface sensitive (detection depth of ~ 5 nm), e.g., the ex-situ characterizations shown in **Figure 2g-h**. However, the FY mode is less surface sensitive (detection depth could be ~ 100 nm). As far as we know, current operando soft X-ray characterizations performed in the literature are typically in FY mode (Nat Catal 7, 1213–1222 (2024)) or partial TEY mode (Nature 587, 408–413 (2020)). It is still very challenging to achieve surface sensitive TEY characterizations for electrocatalytic reactions, due to the coupling of electrons generated from X-ray adsorption (i.e., the electron signal to be detected) and the electrons for the reactions.

In addition, we want to point out that the energy of soft X-rays is much lower than that of hard X-rays, therefore soft X-rays are difficult to penetrate a through thick electrolyte layer. As a result, operando soft X-ray characterization would require a new design of our operando cell, which would likely take years, and therefore it is unlikely to be done timely for this publication. Due to these inconveniences, we hope to have your understanding that we could

not perform the as-suggested experiment, though we ourselves are also interested in operando soft X-ray characterizations in the future.

Comment 7.

I do not fully agree with the authors' assertion that "only a thin surface layer of CoOOH is reconstructed and oxidized into Co(IV) species under OER conditions." To verify this claim, CoOOH samples with different thicknesses should be synthesized, and the relationship between Co valence and sample thickness should be evaluated. If the CoOOH layer is sufficiently thin, would a larger shift in Co valence be observed?

Response: Thank you for your feedback on our discussion about surface oxidation in CoOOH. We fully agree with you that if the CoOOH layer is thinner, a larger shift in Co oxidation state can be expected. Similarly, size dependence of oxidation state change have been well studied by Haase et al. (referred to Nat Energy 7, 765–773 (2022)). Under **alkaline** OER conditions, the average Co oxidation state is $\sim 2.6+$ and $\sim 3.2+$ for the 9 nm and 1 nm CoO_x, respectively. In addition, the change in average oxidation state is also dependent on the catalyst crystallinity. Reported by Prof. Holger Dau and coworkers, an increase of average Co oxidation state from $\sim 2.9+$ to $\sim 3.3+$ is observed in the amorphous Co catalysts under **neutral** OER conditions (referred to J. Am. Chem. Soc. 2019, 141, 2938–2948). However, with the following new experiments and related discussion, we would like to point out it is challenging to perform a size- or thickness-dependent study for Co catalysts under **acidic** conditions, due to dissolution of the catalysts.

As you suggested previously in **Comment 1**, the amorphous Co catalysts (also named cobalt phosphate, i.e., CoPi in the literature) are well studied in neutral environments. We note that both the amorphous CoPi and the crystalline CoOOH are composed of CoO₆ octahedral layers (**Figure R2** shown previously), and a larger oxidation state change is expected in the CoPi if it can be stable in acidic environments.

We deposited the CoPi catalyst on a glassy carbon electrode (**Figure R4a**), following a similar chronoamperometry method reported by Prof. Holger Dau and coworkers (referred to Adv. Energy Mater. 2024, 2403818). After the deposition, the capacitive current for the Co redox processes further increase during the CV measurement, indicating the in-situ generated CoPi catalyst is thickened over the CV cycling (**Figure R4b**). The CVs become quasi-stable after 50 CV cycles. Later, CVs of the in-situ deposited CoPi catalysts was recorded in 0.1 M potassium phosphate (KPi) buffer, 0.05 M H₂SO₄ electrolyte and then back to 0.1 M KPi again (**Figure R4c**). It is obvious that the OER activity of the CoPi catalyst is significantly decreased in a 0.05 M H₂SO₄ electrolyte when compared to that in the 0.1 M KPi buffer. The OER activity

is not recoverable, and the Co redox current density is also significantly decreased when the catalyst is put back into the 0.1 M KPi buffer. With these control experiments, we know that the CoPi suffers from significant Co dissolution in the acidic electrolytes, which can be expected according to our experience. Furthermore, we also compared the OER performance of CoPi with CoOOH in both neutral and acidic electrolytes (**Figure R4d-e**). The CoPi is more active than CoOOH in a neutral electrolyte, which could result from its more abundant active sites in the amorphous structure (as reflected by its larger Co redox current density). However, its activity is significantly lower compared to the CoOOH in an acidic electrolyte, due to its lower stability in acidic environments. Therefore, we want to address that it is important to make Co catalysts more stable before reducing their size or their crystallinity, which remains to be a big challenge in this field and requires long-term efforts that are outside the scope of this publication.

Figure R4. OER performance of the in-situ deposited CoPi catalysts. (a) CA curves for the deposition of CoPi at 1.05 V vs. SHE and (b) CV measurement after the CA deposition in 0.1 M KPi with the presence of 50 mM Co^{2+} , with a scan rate of 100 mV s^{-1} . (c) The CVs of CoPi under different conditions. (d-e) Comparison of CVs for CoOOH and CoPi in (d) 0.1 M KPi buffer and (e) 0.05 M H_2SO_4 electrolyte. The CVs in (c-e) are recorded at a scan rate of 10 mV s^{-1} .

Comment 8.

The relationship between Co valence and the half-edge energy (the energy at half the edge-jump intensity) should be examined. Using $\text{Co}(\text{OH})_2$ as a Co^{2+} reference and CoOOH as a Co^{3+} reference could provide a more quantitative estimate of the Co valence state.

Response: Thank you for this professional comment. As discussed in the response to **Comment 7** above, we note that the average oxidation state under OER conditions in 9 nm CoO_x (~2.6+) is significantly different from that in 1 nm CoO_x (~3.2+, referred to Nat Energy 7, 765–773 (2022)). This is because the XAS characterization is a bulk-averaged technique, and only the Co species in a certain thickness of the surface layer is oxidized under OER conditions. As you pointed out previously, the shift in the Co-K edge decreases with the particle size. However, due to Co dissolution in acidic environments (e.g., the CoPi catalyst), it is also not ideal to use catalysts of very small particle size or low crystallinity. For the CoOOH catalyst herein, we observe an energy shift (i.e., ΔE_{edge}) of ~ 0.15 eV in both alkaline and acidic electrolytes, so the change in the average oxidation state is as little as ~ 0.066 (according to the linear regression: oxidation state change = $\Delta E_{\text{edge}}/2.29$, referred to J. Am. Chem. Soc. 2019, 141, 2938–2948), much less significant than that for the bulk active CoPi catalyst (from ~2.9+ to ~3.3). Such a small change in the average oxidation state will not provide any valuable information about **surface reconstruction** to the reader, except to mislead. Therefore, we want to have your understanding on our insistence of using ΔE_{edge} as the parameter, instead of average oxidation state, when elucidating trends in the interfacial Co oxidation.

Supplementary Figure 10. Shift of adsorption energy at Co-K edge extracted by different methods for CP measurement in acidic electrolyte. (a) E_{edge} determined from the integral method are compared to the energies at $\mu = 0.5$ (i.e., at half of the edge-jump intensity) and $\mu = 0.6$ of the normalized intensity, which are denoted as $E_{0.5}$ and $E_{0.6}$, respectively. (b) Comparison of the shifts of adsorption energy extracted by different methods, i.e., ΔE_{edge} , $\Delta E_{0.5}$ and $\Delta E_{0.6}$, made by subtracting the E_{edge} at 0s.

Furthermore, we note that both the energy at half of the edge-jump intensity (proposed by you) and the integral method used in this manuscript are both commonly applied to extract the edge position of the XAS spectra in the literature, as summarized in the **Review** by Prof. Hao Ming Chen and coworkers (Nat Commun 14, 6576 (2023)). With regards to using ΔE_{edge}

as a parameter, we will show that these two methods are equivalent, by doing new analysis on the operando XAS data collected during the CP measurement in an acidic electrolyte as the example (the XAS spectra are shown in **Figure 1c-d** of the main text). As shown in the **New Supplementary Figure 10**, the E_{edge} values determined from the integral method ($E_{\text{edge}} = \frac{1}{\mu_2 - \mu_1} \int_{\mu_1}^{\mu_2} E(\mu) d\mu$

, $\mu_1 = 0.2$ and $\mu_2 = 1$) are located between the energies at $\mu = 0.5$ (i.e., at half of the edge-jump intensity) and $\mu = 0.6$ of the normalized intensity, which are denoted as $E_{0.5}$ and $E_{0.6}$, respectively. As E_{edge} is an averaged value when obtained by the integral method, its plot as a function of time is smoother compared to those for $E_{0.5}$ and $E_{0.6}$. When the shifts in the adsorption energy at $t > 0$ s are extracted by subtracting the E_{edge} at 0 s, we find the ΔE_{edge} is closely overlapping with the $\Delta E_{0.5}$ and $\Delta E_{0.6}$, indicating that the shift in energy is not influenced by the different methods used to extract the adsorption energy at the Co K edge. To make it clear to the readers, we have added a comment on page 7 line 6 of the main text as follows:

“We note that the ΔE_{edge} determined by the integral method is equivalent to the conventional method of using the energy at half the edge-jump intensity (**Supplementary Figure 10**).”

To further prove the good match between ΔE_{edge} , $\Delta E_{0.5}$ and $\Delta E_{0.6}$, similar analysis has also been done to the operando XAS data during the CV measurement (shown in **Fig. 5a** of the main text). The results are also provided for your reference, as shown in **Figure R5** below. Generally, we demonstrate that the XAS spectra interpretation will not be affected by the method used to determine the adsorption energy of Co-K edge. However, the small ΔE_{edge} clearly indicates that only a small proportion of surface Co atoms are oxidized under OER conditions. To avoid misleading the readers, we hope you can agree that it is more ideal to keep using ΔE_{edge} as the parameter to semi-quantitatively reflect the interfacial Co oxidation state,

instead of converting into average Co oxidation state.

Figure R5. Shift of adsorption energy at Co-K edge extracted by different methods for CV measurement. (a) E_{edge} determined from the integral method are compared to the energies at

$\mu = 0.5$ (i.e., at half of the edge-jump intensity) and $\mu = 0.6$ of the normalized intensity, which are denoted as $E_{0.5}$ and $E_{0.6}$, respectively. (b) Comparison of the shifts of adsorption energy extracted by different methods, i.e., ΔE_{edge} , $\Delta E_{0.5}$ and $\Delta E_{0.6}$, by subtracting the subtracting the E_{edge} at 0s.

Comment 9.

The photograph of the flow cell in Supplementary Figure 5 is unclear. A schematic diagram should be provided instead. Additionally, the significance of this measurement setup should be clarified, as it appears to be a standard operando XAFS cell.

Response: Thank you for the suggestions. We have provided the detailed schematic diagram of the flow cell in the paper previously published by our group (ACS Catal. 2022, 12, 17, 10727–10741). To avoid duplicates in the publications but also to address your concerns, we now have added pictures of the three main PEEK components of the flow cell in new **Supplementary Figure 7**, as shown below. The paper has also been cited in the caption to notify readers who are interested in the detailed cell design. In addition, we have also clarified this information in **Methods** on page 21 line 20:

“The operando experiments were performed in a spectro-electrochemical flow cell, with the cell design detailed in previous works^{27, 62}.”

Supplementary Figure 7. Flow cell. Photos of (a) the flow cell used for operando XAS characterizations and (b) the main PEEK cell components. The schematic diagram and other details about this operando flow cell can be found in previous publications^{1, 2}

Response Letter for

Directly synthesized cobalt oxyhydroxide as an oxygen evolution catalyst in proton exchange membrane water electrolyzers (NCOMMS-25-06391B)

Dear Reviewers,

We appreciate your further input in reviewing our revised manuscripts. We are delighted that most of the concerns have been adequately addressed. We have tried our best to resolve the remaining issues in this second revision. The point-by-point responses to each comment/suggestion are outlined in the following text. The revisions made in the main text and in the Supplementary Information are highlighted in yellow.

Reviewer #1

All concerns related to my questions have been addressed appropriately. Therefore, I recommend acceptance of this manuscript for publication in Nature Communications.

Response: Thank you for your time and effort in reviewing our manuscript.

Reviewer #2

In this revised version, the authors have addressed most of the questions effectively. However, some issues remain unanswered due to a lack of experimental evidence. Additionally, certain details could potentially mislead readers or create ambiguity. In my opinion, the scientific rigor and robustness of this manuscript still require further improvement. Therefore, I cannot recommend this version for publication in its current state.

Response: Thank you for your recognition of the effort we made in last revision, and your further diligent review of the revised manuscripts. We appreciate your comments, which we would like to address to further improve the rigor of the related discussion.

Comment 1

In response to Comment 3, the authors attributed the absence of FT-IR experiments to instrument malfunction—an explanation that appears unsubstantiated. Furthermore, the DEMS results failed to demonstrate clear ^{18}O detection signals due to reportedly low intensity. Does this imply that the material in this paper does not follow the OPM pathway? Given the extensive use of DEMS (J. Am. Chem. Soc. 2023, 145, 7829–7836 (2023); Adv. Mater. 37,

2411709 (2024).) in validating OPM mechanisms in prior studies, the authors cannot dismiss this issue perfunctorily. While the OPM mechanism is relatively novel compared to AEM and LOM, and could significantly enhance the scientific depth of this work, the current lack of compelling evidence weakens its credibility.

Response: Thank you for your insightful comment on the OER mechanism discussion. It is indeed one of the most attractive and challenging research topics for OER research. We appreciate your recommendations to use in-situ FT-IR and DEMS to provide more insightful details to support the mechanistic discussion. These characterizations could have been performed if they were convenient. As a research group with a strong background in operando techniques, we began designing our own FT-IR cell two years ago. We have had to slow down the process since we are still waiting for the new setup to arrive. In addition, there are experimental challenges that so far we could not resolve. For example, the Si crystal used for FTIR is not stable in alkaline environments, and its instability precludes the correct interpretation of the data. We have tried different crystals, but without obtaining enough signal to noise ratio in operando mode. While the methods used for the deposition of Au layer on the crystals also affects the stability in the acidic environments. Therefore, we still need to further optimize the cell design to enable reliable and stable data collection. We hope to have your understanding that the requested FT-IR measurements cannot be performed within the scope of this publication.

We also thank you for the kind recommendation of these two papers (J. Am. Chem. Soc. 2023, 145, 7829–7836 (2023); Adv. Mater. 37, 2411709 (2024).) which use DEMS to track the participation of surface oxygen species in the OER cycle. Note that, in J. Am. Chem. Soc. 2023, 145, 7829–7836 (2023), DEMS is indeed used to *differentiate the OPM pathway from AEM pathway*; however, in Adv. Mater. 37, 2411709 (2024), DEMS is actually used to *differentiate the lattice oxygen mechanism (LOM) pathway from the AEM pathway*.

Figure R1. Comparison of key O-O coupling in the oxide path mechanism (OPM) versus lattice oxygen mechanism (LOM). The O atoms axially adsorbed onto of the surface Co sites are in **yellow**, while the O atoms in the lattice are in **blue**.

Similarly, as discussed in our response in the first revision, DEMS has been applied to support the LOM pathway in Nat. Chem. 9, 457-465 (2017), but the OPM pathway in Nat

Catal 4, 1012–1023 (2021). The difference in O-O coupling for LOM versus OPM is depicted in Figure R1. To the best of our knowledge, it is still challenging to use DEMS to differentiate these two pathways at the current stage, since DEMS is not a technique that enables a direct probe of surface processes. In particular, Co-based catalysts are known to undergo more significant surface reconstruction than Ir/Ru-based catalysts. This suggests that the O species on the surface will rearrange under OER conditions, making it more difficult to differentiate the OPM from the LOM pathway.

Another important fact is that, similar to other Co-based catalysts, the CoOOH studied here is meta-stable. As shown in our Supplementary Table 4, its stability number is calculated to be ~700. Its stability number is 1-2 orders of magnitude lower than that of Ru-based catalysts and 2-3 orders of magnitude lower than that of Ir-based catalysts (details in Figure 6 of ACS Catal. 2023, 13, 14058–14069). Recently, it is also suggested that Co-based catalysts suffer from transient dissolution (J. Am. Chem. Soc. 2025, 147, 4, 3517–3528). Specifically, significant dissolution happens at low applied potentials before the onset of the OER. The dynamic Co dissolution, accompanied by the loss of lattice oxygen dissolved into the electrolyte, would further challenge proper labelling of the CoOOH surface with an ^{18}O isotope, under reaction conditions, as well as reliable interpretation of DEMS results.

In this work, the OPM pathway is proposed for CoOOH based on previous papers investigating Co-based catalysts (Science 384, 1373-1380 (2024); J. Am. Chem. Soc. 2023, 145, 14, 7829–7836). To avoid misleading the readers, we have already reframed the discussion in the “OER mechanism” section and clearly pointed out the limitations of our analyses (starting from page 14 line 9 of the main text). We appreciate your recommendation to provide more experimental evidence to clarify the reaction mechanism. We strongly agree with you that this topic is extremely important for OER research. However, we hope to have your understanding that DEMS cannot ideally achieve the goal of clarifying the OPM and LOM pathways, which is still challenging and under discussion in the literature.

Comment 2

The authors mentioned that “There is only one discernable redox pair observed in the acidic electrolyte, in contrast to the two pairs of redox peaks that are assigned to Co II/III and Co III/IV redox in alkaline and quasi-neutral electrolytes”. In fact, at pH=7 in Figure 2a, no distinct oxidation peak was observed. At pH=1, a reduction peak appeared around a potential of 1.67 V. Therefore, the CV patterns under neutral and acidic conditions are similar, which seems to contradict the author's claims.

Response: Thank you for your careful checking of our discussion about pH-dependent Co redox properties. To improve the rigor of our discussion, we have reconstructed the related sentences on page 5 line 10 of the main text, as follows.

“In an alkaline electrolyte (pH ~ 13), two pairs of redox peaks are assigned to $\text{Co}^{\text{II/III}}$ and $\text{Co}^{\text{III/IV}}$ redox processes at the low and high applied potentials, respectively.²² In a neutral electrolyte (pH ~ 7), the $\text{Co}^{\text{II/IV}}$ reduction peak is still identifiable, but the $\text{Co}^{\text{III/IV}}$ oxidation peak is not. Furthermore, there is only one discernable redox pair observed in the acidic electrolyte (pH ~ 1).”

Comment 3

How do you define the corresponding formal redox potential (denoted as $V^\circ(\text{Co}^{\text{II/III}})$)? In Supplementary Figure 2c, the max potential of oxidation peak is close to 1.6 V, which seems to contradict the value in Fig. 2b.

Response: Thank you for your comments. $V^\circ(\text{Co}^{\text{II/III}})$ is the average of the oxidation and reduction peak potentials, therefore it is reasonable to be less than the potential of the oxidation peak. We provided details of the calculation of the formal redox potential on page 5 line 23 of the main text:

“The $\text{Co}^{\text{II/III}}$ redox formal potentials ($V^\circ(\text{Co}^{\text{II/III}}) = (V_a + V_c)/2$, where V_a and V_c are the potentials of the anodic and cathodic redox peaks, respectively) were extracted from the CVs at different scan rates (Supplementary Figure 2), as summarized in Figure 2b. The pH dependence of $V^\circ(\text{Co}^{\text{II/III}})$ is clearly evident.”

Comment 4

The Mott-Schottky analysis the author used should be clarified in details (e.g. measurement methods, equation and parameters).

Response: Thank you for bringing this issue to our attention. We have added the details about the Mott-Schottky analysis following Supplementary Figure 12 on page 17 of the Supplementary Information, as follows.

“**Discussion:** The electrochemical impedance spectroscopy data (Supplementary Figure 11) were collected with a Staircase Potentiostatic Electrochemical Impedance Spectroscopy (Mott-Schottky) technique using the Biologic VMP-300 software. The frequency was changed from 100 kHz to 1 Hz. The potential windows for the measurement in 0.05 M H_2SO_4 and 0.1 M KOH are 1.0 to 1.65 V vs. RHE and 0.8 to 1.7 V vs. RHE, respectively. The $1/C_s^2$ at different frequencies can directly be extracted using this software, and were plotted against the applied potentials to derive Mott-Schottky plots³ in Supplementary Figure 12a-b. The flat band

potential is determined from the x-intercept of the linear fit, as shown in **Supplementary Figure 12c.**"

The new Ref.3 in the Supplementary Information (Curr. Sep. 17, 87-92 (1998)) about Mott-Schottky analysis for semiconductors is cited to support the discussion.

Comment 5

This work appears to bear similarity to another paper "Oxidation of interfacial cobalt controls the pH dependence of the oxygen evolution reaction" published in Nature Chemistry, with research object, study methods, and some conclusions seemingly overlapping. It creates confusion regarding the novelty of the findings presented in this manuscript.

Response: Thank you for your recognition of our Nat Chem paper (cited as ref. 24, Nat. Chem. 17, 856–864 (2025)) published online recently. We will address your concerns about i) the research objective, ii) the study methods, and iii) the conclusions in the following.

i) Research objective. In our latest Nat Chem paper, we used a commercial CoO_x catalyst, mainly composed of Co₃O₄ spinel oxide, to reveal that interfacial Co oxidation controls the pH-dependence of OER, which has broad implications for the electrocatalysis community. In the current manuscript, we proved that CoOOH, a structure close to the reconstructed active site of Co-based catalysts, can actually catalyze the OER in acidic environments, **which has not yet been adequately explored in the literature**. Even though the studied catalysts are both Co-based materials, their reconstructed surfaces formed in situ are different, as detailed in the previous paper by Prof. Peter Strasser and coworkers (cited as ref.16, Nat Catal 1, 711–719 (2018)). However, we note that surface reconstruction is both structure-dependent and pH-dependent. Using commercial CoO_x in our latest paper, and CoOOH herein, we have tried to find answers for two remaining questions in the field: **1) what controls the pH dependence of the OER and 2) whether CoOOH can directly catalyze OER in acidic environments or not,** which are intrinsically different.

ii) Study methods. In the past ten years, our group has developed our key strength of using operando hXAS characterizations to uncover surface reconstruction processes occurring during the OER and beyond. We hope to have your understanding that the novelty of this work is not about the techniques used but the observations from the experiments. Moreover, in this work, to provide deeper mechanistic insights, we have performed extra experiments that have not been done in our latest paper, e.g., an isotope substitution study, a temperature dependent study and density function theory calculations, etc. Specifically, one of the main research objectives of this work is to **verify the performance of CoOOH in a real**

PEM water electrolyzer, which to the best of our knowledge has not yet been done in the literature.

iii) Conclusions. As mentioned above, we did apply operando hXAS in both our latest Nat Chem paper and this work. However, we would like to draw your attention to the resulting observations, which are interestingly different, as compared in **Figure R2** below. Note that, in the low-current region (i.e., $\log(i) \leq -1$), the surface charge is dominating; while in the high-current region (i.e., $\log(i) > -1$), the OER is dominating. Therefore, the ΔE_{edge} seen at $\log(i) = -1$ can serve as an indicator for the abundance of oxidized Co species formed before the OER onset.

Figure R2. Comparison of interfacial Co oxidation (indicated by the shift in the Co-K edge, i.e., ΔE_{edge}) between (a) commercial CoO_x (Figure 4a in Nat. Chem. 17, 856–864 (2025)) and (b) the CoOOH used in this work (Figure 3b in the main text of this work).

In commercial CoO_x, its surface will first reconstruct into a CoOOH-like structure, and then be further oxidized to form Co^{IV} active centers. We found that the reconstruction into a CoOOH-like structure is pH-dependent, as indicated by larger ΔE_{edge} at $\log(i) = -1$ seen in an alkaline electrolyte when compared to an acidic electrolyte (**Figure R2a**). In fact, this observation arises because some of the Co^{II} species are difficult to oxidize into Co^{III} species in acidic electrolytes (cited as ref. 6, Nat Commun 15, 3067 (2024)). In comparison, the ΔE_{edge} at $\log(i) = -1$ seen in CoOOH in this work is comparable in both alkaline and acidic electrolyte (**Figure R2b**), indicating that the abundance of redox-active Co species at the interface is unaffected by the pH environment. To clearly highlight these differences, we have added the following discussion on page 9 line 31 of the main text.

“These results differ from those found in a commercial CoO_x (ref.²⁴), in which the redox charge and the extent of changes in the Co oxidation transformation are pH-dependent. This is likely

due to the fact that $\text{Co}^{\text{II/III}}$ oxidation is less favorable in an acidic electrolyte than in an alkaline electrolyte⁶.”

We hope these revisions and clarifications can adequately address your concerns. Thank you for your time and effort on the second-round review of our manuscripts.

Reviewer #3

The authors have thoroughly addressed my comments and provided appropriate clarifications supported by additional data and thoughtful discussion. The manuscript has been substantially improved and now presents a clearer and more comprehensive understanding of the OER activity and mechanism of CoOOH under acidic conditions. Therefore, I support the acceptance of this manuscript for publication in Nature Communications.

Response: Thank you for your thoughtful comments/suggestions in last revision that have helped to improve our manuscript.